


# Inputs and processes affecting the distribution of particulate iron in the North Atlantic along the GEOVIDE (GEOTRACES GA01) section

Arthur Gourain[1,2], Hélène Planquette[1], Marie Cheize[1,3], Nolwenn Lemaitre[1,4], Jan-Lukas Menzel Barraqueta[5], Rachel Shelley[1,6], Pascale Lherminier[7] and Géraldine Sarthou[1]

1-UMR 6539/LEMAR/IUEM, Technopôle Brest Iroise, Place Nicolas Copernic, 29280 Plouzané, France

2- now at Ocean Sciences Department, School of Environmental Sciences, University of Liverpool, Liverpool, L69 3GP, United Kingdom

3- now at Ifremer, Centre de Brest, Géosciences Marines, Laboratoire des Cycles Géochimiques (LCG), 29280 Plouzané, France

4- now at Department of Earth Sciences, Institute of Geochemistry and Petrology, ETH-Zürich, Zürich, Switzerland

5- GEOMAR, Helmholtz Centre for Ocean Research Kiel, Wischhofstraße 1-3, 24148 Kiel, Germany

6- now at Earth, Ocean and Atmospheric Science, Florida State University, Tallahassee, Florida, 32310, USA

7- Ifremer, LPO, UMR 6523 CNRS/Ifremer/IRD/UBO, Ifremer Centre de Brest, CS 10070, Plouzané, France

*Correspondence to: helene.planquette@univ-brest.fr*

**Abstract**

The GEOVIDE cruise (May-June 2014, R/V Pourquoi Pas?) aimed to provide a better understanding on trace metal biogeochemical cycles in the North Atlantic. As particles play a key role in the global biogeochemical cycle of trace elements in the ocean, we discuss the distribution of particulate iron (PFe), in light of particulate aluminium (PAl), manganese (PMn) and phosphorus (PP) distributions. Overall, 32 full vertical profiles were collected for trace metal analyses, representing more than 500 samples. This resolution provides a solid basis for assessing concentration distributions, elemental ratios, size-fractionation, or adsorptive scavenging processes in key areas of the thermohaline circulation. Total particulate iron (PFe) concentrations ranged from as low as 9 pmol L$^{-1}$ in surface Labrador Sea waters to 304 nmol L$^{-1}$ near the Iberian margin, while median PFe concentrations of 1.15 nmol L$^{-1}$ were measured over the sub-euphotic ocean interior.

At most stations over the Western, the relative concentrations of total PFe and aluminium (PAl) showed the near-ubiquitous influence of crustal particles in the water column. Overall, the lithogenic component explained more than 87% of PFe variance along the section. Within the Irminger and Labrador basins, the formation of biogenic particles led to an increase of the PFe/PAl ratio (up to 0.7 mol mol$^{-1}$) compared to the continental crust ratio (0.21 mol mol$^{-1}$), Margins provide important quantities of particulate trace elements (up to 10 nmol L$^{-1}$ of PFe) to the open ocean, and in the case of the Iberian margin, advection of PFe was visible more than 250km



away from the margin. Additionally, several benthic nepheloid layers spreading over 200m above the seafloor
were encountered along the transect, especially in the Icelandic, Irminger and Labrador basins, delivering
particles with high PFe content, up to 89 nmol L$^{-1}$ of PFe. Finally, remineralisation processes are also discussed,
and showed different patterns among basins and elements.
**1. Introduction**
Particles play a key role in the ocean where they drive the residence time of most elements (Jeandel et al., 2015),
and strongly influence the global biogeochemistry of macro and micro-nutrients including iron (Milne et al.,
2017). In the surface ocean, biological activity produces biogenic suspended matter through planktonic
organisms, while atmospheric deposition (Baker et al., 2013; Jickells et al., 2005), riverine discharge (Aguilar-
Islas et al., 2013; Berger et al., 2008; Ussher et al., 2004) or ice-melting (Hawkings et al., 2014; Lannuzel et al.,
2011, 2014) bring mostly lithogenic derived particles to surface waters. These particulate inputs highly vary,
both spatially and seasonally, around the world's oceans. At depth, benthic and shelf sediment resuspension
(e.g. Aguilar-Islas et al., 2013; Cullen et al., 2009; Elrod et al., 2004; Fitzwater et al., 2000; Hwang et al., 2010;
Lam et al., 2015; Lam and Bishop, 2008; McCave and Hall, 2002), and hydrothermal activity (Elderfield and
Schultz, 1996; Lam et al., 2012; Tagliabue et al., 2010, 2017; Trefry et al., 1985), provides important amounts
of particles to the water column.  Moreover, authigenic particles can be produced *in-situ* by aggregation of
colloids (Bergquist et al., 2007) or oxidation processes (Bishop and Fleisher, 1987; Collier and Edmond, 1984).
Thus, oceanic particles result from a complex combination of these different sources and processes (Lam et al.,

2015).

Particles represent the main part of the total iron pool in the upper water column (Radic et al., 2011), and
strongly interact with the dissolved pool (e.g. Ellwood et al., 2014). Indeed, dissolved iron can be scavenged
onto particles (Gerringa et al., 2015; Rijkenberg et al., 2014), incorporated into biogenic particles (Berger et al.,
2008) or remineralised (Dehairs et al., 2008; Sarthou et al., 2008). Interestingly, the concept of "reversible
scavenging" (i.e. release at depth of dissolved iron previously scavenged onto particles) has been advocated
recently (Dutay et al., 2015; Jeandel and Oelkers, 2015; Labatut et al., 2014), while other studies reveal distinct
dissolution processes (e.g. Oelkers et al., 2012; Cheize et al., submitted to Chemical Geology). Slow dissolution
of particulate iron at margins has also been evoked as a continuous fertilizer of primary production and should
be considered as a source of dissolved iron (e.g. Jeandel et al., 2011; Jeandel and Oelkers, 2015; Lam and
Bishop, 2008). Within or below the mixed layer, the rates of regeneration processes can also impact the
bioavailable pool of iron, among other trace metals (e.g. Ellwood et al., 2014; Nuester et al., 2014). However,
the rates of these processes are not yet fully constrained. The study of particulate iron is thus essential to better
constrain the global biogeochemical cycle of iron in the ocean. This subject received a growing interest over the
last 10 years in particular (e.g. Bishop and Biscaye, 1982; Collier and Edmond, 1984; Frew et al., 2006; Lam et
al., 2012; Milne et al., 2017; Planquette et al., 2011, 2013; Sherrell et al., 1998) and, to our knowledge, only two
have been performed at an ocean-wide scale and published so far: the GA03 GEOTRACES North Atlantic
Zonal Transect (Lam et al., 2015; Ohnemus and Lam, 2015) and the GP16 GEOTRACES Pacific Transect (Lam
et al., 2017; Lee et al., 2017).
In this context, this paper presents the particulate iron distribution in the North Atlantic Ocean, along the
GEOTRACES GA01 section (GEOVIDE), and discusses the various sources and processes affecting its



distribution, using the distribution of other trace elements, more particularly particulate aluminium, phosphorus
or manganese, to further our understanding of this important pool of iron.

**2. Methods**

*2.1. Study area*

Particulate samples were collected at 32 stations during the GEOVIDE (GEOTRACES GA01 section) campaign
between May and June 2014 aboard the R/V *Pourquoi Pas?* in the North Atlantic. The sampling spanned
several biogeochemical provinces (Figure 1): the Iberian margin (IM, Stations 2, 1 and 4), the Iberian Abyssal
Plain (IAP, Stations 11 to 17), the Western European Basin (WEB, Station 19 to Station 29), the Icelandic Basin
(IcB, Stations 32 to 36), above the Reykjanes Ridge (RR, Station 38), the Irminger Basin (IrB, Stations 40 to
60), the Greenland shelf (GS, Stations 53 and 61), the Labrador Basin (LB, Stations 63 to 77) and finally the
Newfoundland shelf (NS, Station 78) (Figure 1).
The North Atlantic is characterized by a complex circulation (briefly described in section 2.1 and in detail by
Zunino et al. (2017) and García-Ibáñez et al. (2015) and is one of the most productive regions of the global
ocean (Martin et al., 1993; Sanders et al., 2014), with a complex phytoplankton community structure composed
of diverse taxa (Tonnard et al., in prep.).


*2.2. Sampling*

Samples were collected using the French GEOTRACES clean rosette, equipped with twenty-two 12L GO-FLO
bottles (two bottles were leaking and were never deployed during the cruise). GO-FLO bottles were initially
cleaned in the home laboratory (LEMAR ) following the GEOTRACES procedures (Cutter and Bruland, 2012).
The rosette was deployed on a 6mm Kevlar cable with a dedicated, custom-designed clean winch. Immediately
after recovery, the GO-FLO bottles were individually covered at each end with plastic bags to minimize
contamination. They were then transferred into a clean container (class-100) for sampling, and the filters
processed under a laminar flow unit. On each cast, nutrient and/or salinity samples were taken to check potential
leakage of the GO-FLO bottles.
Prior to filtration, the GO-FLO bottles were shaken three times, as recommended in the GEOTRACES
cookbook to avoid settling of particles in the lower part of the bottle. GO-FLO bottles were pressurized to <8 psi
with 0.2 µm filtered dinitrogen ($N_2$, Air Liquide). Seawater was then filtered directly through paired filters (Pall
Gelman Supor[TM] 0.45 µm polyetersulfone, and Millipore mixed ester cellulose MF 5 µm) mounted in Swinnex
polypropylene filter holders (Millipore), following Planquette and Sherrell (2012) inside the clean container.
Filtration was operated until the bottle was empty or until the filter clogged; volume filtered ranged from 2
litters for surface samples to 11L within the water column. Filters were cleaned following the protocol described
in Planquette and Sherrell (2012) and kept in acid-cleaned 1 L LDPE bottles (Nalgene) filled with ultrapure
water (Milli-Q, resistivity of 18.2 MΩ cm-[1]) until use. All filters were 25 mm diameter in order to optimize
signal over the filter blank except at the surface depth where 47 mm diameter filters mounted on acid-cleaned
polysulfone filter holders (Nalgene[TM]) were used. After filtration, filter holders were disconnected from the GO-
FLO bottles and a gentle vacuum was applied using a syringe in order to remove any residual water under a



laminar flow hood. Filters were then removed from the filter holders with plastic tweezers that were rinsed with
Milli-Q between samples. Most of the remaining seawater was 'sipped' by capillary action, when placing the
non-sampled side of the filter onto a clean 47 mm supor filter. Then, each filter pair was placed in an acid-
cleaned polystyrene Petri slide (Millipore), double bagged, and finally stored at -20°C until analysis at LEMAR.
Between casts, filter holders were thoroughly rinsed with Milli-Q, placed in an acid bath (5% HCl) for 24 hours,
then rinsed with Milli-Q.
At each station, process blanks were collected as follows: 2L of a deep (1000 m) and a shallow (40 m) seawater
samples were first filtered through a 0.2 μm pore size capsule filter (Pall Gelman Acropak 200) mounted on the
outlet of the GO-FLO bottle before to pass through the particle sampling filter, which was attached directly to
the swinnex filter holder.

*2.3. Analytical methods*
Back in the home laboratory, sample handling was performed inside a clean room (Class 100). All solutions
were prepared using ultrapure water (Milli-Q) and all plasticware had been acid-cleaned before use. Frozen
filters, collected within the mixed layer depth or within nepheloid layers, were first cut in half using a ceramic
blade: one filter half was dedicated to total digestion (see below), while the other half was archived at -20°C for
SEM analyses or acid leaching of "labile" metals (Berger et al., 2008; to be published separately).
Filters were digested following the method described in Planquette and Sherrell (2012). Filter were placed on
the inner wall of acid-clean 15mL PFA vials (Savillex$^{TM}$), and 2 mL of a solution containing 2.9 mol L$^{-1}$
hydrofluoric acid (HF, suprapur grade, Merck) and 8 mol L$^{-1}$ nitric acid (HNO$_3$, Ultrapur grade, Merck) was
added to each vial. Vials were then closed and refluxed at 130°C on a hot plate for 4 hours. After cooling, the
digest solution was evaporated at 110°C until near dryness. Then, 400 μL of concentrated HNO$_3$ (Ultrapur
grade, Merck) was added, and the solution was re-evaporated at 110°C. Finally, the obtained residue was
dissolved with 3mL of a 0.8 mol L$^{-1}$ HNO$_3$ (Ultrapure grade, Merck). This archive solution was transferred to an
acid cleaned 15 mL polypropylene centrifuge tube (Corning®) and stored at 4°C until analyses.

All analyses were performed on a sector field inductively coupled plasma mass spectrometer (SF-ICP-MS
Element2, Thermo-Fisher Scientific). Samples were diluted by a factor of 7 on the day of analysis in acid-
washed 13 mm (outer diameter) rounded bottom, polypropylene centrifuge tubes (VWR) with 0.8 mol L$^{-1}$ HNO$_3$
(Ultrapur grade, Merck) spiked with 1μg L$^{-1}$ of Indium ($^{115}$In) solution in order to monitor the instrument drift.
Samples were introduced with a PFA-ST nebulizer connected to a quartz cyclonic spray chamber (Elemental
Scientific Incorporated, Omaha, NE) via a modified SC-Fast introduction system consisting of an SC-2
autosampler, a six-port valve and a vacuum-rinsing pump. The autosampler was contained under a HEPA
filtered unit (Elemental Scientific). Two 6-points, matrix-matched multi-element standard curves with
concentrations bracketing the range of the samples were run at the beginning, the middle and the end of each
analytical run. Analytical replicates were made every 10 samples, while accuracy was determined by performing
digestions of the certified reference material BCR-414 (plankton, Community Bureau of Reference,
Commission of the European Communities), PACS-3 and MESS-4 (marine sediments, National Research
Council Canada), following the same protocol as for samples. Recoveries were typically within 10% of the
certified values (and within the error of the data, taken from replicate measurements, Table 1).





Once all data were normalized to an $^{115}$In internal standard and quantified using an external standard curve, the
dilution factor of the total digestion was accounted for.  Obtained element concentrations per filter (pmol/filter)
were then corrected by the process blanks described above. Finally, pmol/filter values were divided by the
volume of water filtered through stacked filters.
Total concentrations (sum of small size fraction (0.45-5 μm) and large (>5 μm) size fraction) of particulate trace
elements are reported in Table S1 (supplementary data).

*2.4. Ancillary data:*
Particulate barium (Ba) concentrations were determined in samples collected using a standard CTD rosette
equipped with 12 L Niskin bottles. Typically, 18 samples were collected at each station within the first 1000 m.
Details on analytical procedures are given in Lemaitre et al. (in press, 2018a). Briefly, particulate biogenic
Barium, or excess Barium (Ba$_{xs}$), were calculated by subtracting the particulate lithogenic barium (PBa-litho)
from the total particulate barium (PBa). The PBa-litho was determined by multiplying the particulate aluminium
(PAl) concentration by the upper continental crust (UCC) Ba: Al molar ratio (0.00135 mol mol$^{-1}$; Taylor and
Mclennan, 1985). Potential temperature (θ), salinity (S), and transmissometry data were retrieved from the CTD
sensors (CTD SBE911 equipped with a SBE43).

**3. Results**
*3.1. Hydrography and biological setting*
Here, we briefly describe the hydrography encountered during the GEOVIDE section (Figure 2), as a thorough
description is available in García-Ibáñez et al. (2015). The warm and salty Mediterranean Water (MW, S=36.50,
θ°=11.7°C) was sampled between 600 and 1700 m in the Iberian Abyssal Plain (IAP). MW resulted from the
mixing between the Mediterranean Overflow Water plume coming from the Mediterranean Sea and local
waters. Surface water above the Iberian Shelf was characterised by low salinity (S=34.95) at station 2 and 4
compared to surrounding water masses. Close to the floor of the Iberian Abyssal Basin, the North East Atlantic
Deep Water (NEADW, S=34.89, θ°=2.0°C) spread southward. The East North Atlantic Central Water
(ENACW, S>35.60, θ°>12.3°C) was the warmest water mass of the transect and was observed in the subsurface
layer of the Western European Basin and Iberian Abyssal Plain. An old Labrador Sea Water (LSW, S=34.87,
θ°=3. 0°C) flowed inside the Western European and Icelandic Basins, between 1000 and 2500m depth. In the
Icelandic Basin, below the old LSW, the Iceland-Scotland Overflow Water (ISOW, S=34.98, θ°=2.6°C) spread
along the Reykjanes Ridge slope. This cold water, originating from the Arctic, led to the formation of NEADW
after mixing with surrounding waters. North Atlantic hydrography was impacted by the northward flowing of
the North Atlantic Current (NAC), which carried up warm and salty waters from the subtropical area. When
NAC crossed the Mid-Atlantic ridge through the Charlie-Gibbs Fracture Zone (CGFZ), it created the Subpolar
Mode Water (SPMW). The recirculation of SPMW inside the Icelandic and Irminger Basins led to the formation
of regional modal waters: the Iceland Subpolar Mode Water (IcSPMW, S=35.2, θ°=8.0°C) and the Irminger





Subpolar Mode Water (IrSPMW, S=35.01, $\theta°$=5.0°C) respectively. IcSPMW was a relatively warm water mass
with potential temperature up to 7°C (García-Ibáñez et al., 2015). Another branch of the NAC mixed with
Labrador Current waters to form the relatively fresh SubArctic Intermediate Water (SAIW, S=<34.8,
4.5°C<$\theta°$<6°C). The Irminger Basin is a really complex area with a multitude of water masses. In the middle of
the basin, an old LSW, formed one year before (Straneo et al., 2003), spread between 500 and 1200 m depth.
Close to the bottom, the Denmark Strait Overflow Water (DSOW, S=34.91) flowed across the basin. Greenland
coastal waters were characterised by low salinity values, down to S=33. The strong East Greenland Current
(EGC) flowed southward along the Greenland shelf in the Irminger Basin. When reaching the southern tip of
Greenland, this current entered the Labrador Basin along the west coast of Greenland and followed the outskirts
of the basin until the Newfoundland shelf. In the Labrador Basin, the deep convection of SPMW at 2000 m  was
involved in the formation of the LSW (S=34.9, $\theta°$C=3.0°C) (García-Ibáñez et al., 2015; Yashayaev and Loder,
2009). Above the Newfoundland Shelf, surface waters were affected by discharge from rivers and ice-melting
and characterised by extreme low salinity for open ocean waters, below 32 in the first 15 meters.

During GEOVIDE, diatoms and type 6-haptophytes dominated the bloom close to the IM, while type-6-
haptophytes and dinophytes were dominant in the WEB province (Tonnard et al., in prep.). The IB bloom was
dominated by type-6-haptophytes and the IrB was dominated by diatoms. GS and NS coastal stations were
almost exclusively composed of large diatoms. Finally, the LB was dominated by diatoms and type 6 and 8-
haptophytes. The NS, LB, GS and IrB provinces (stations 44 to 77) were sampled just after the bloom peak. The
LB was characterized by an intense particulate organic carbon (POC) export and high remineralization activity
(Lemaitre et al., 2018a).  In contrast, low remineralization fluxes and high POC exports were determined within
the IB and WEB provinces, where the bloom was still active (Lemaitre et al, 2018a, b).
*3.2.  Section overview*
Total particulate iron (PFe), aluminium (PAl), manganese (PMn) and phosphorus (PP) concentrations spanned a
large range of concentrations from below detection to 304, 1544, 21.5, 3.5 and 402 nmol L$^{-1}$ respectively.
Moreover, PFe, PAl, and PMn were predominantly found (>90%) in particles larger than 5 µm, except in
surface waters, where 20% of PFe, 30% of PP, 35% of PAl and up to 60% of PMn were hosted by smaller
particles (0.45-5 µm). The ranges of concentrations are comparable to other studies recently published (Table 2).
Data are shown in Figure 3.

*3.3. Iberian Margin (stations 1 to 4)*
The Iberian margin was characterised by low beam transmissometry values at station 2 (88% at 140 m)
suggesting significant particle concentrations. Particulate iron concentrations varied between 0.02 nmol L$^{-1}$ (20
m) to 304 nmol L$^{-1}$ (138 m) in this area. Above the Iberian Shelf, high PFe concentrations were measured in
surface (Station 2, 2.53 nmol L$^{-1}$); then, on the shelf break, surface concentrations dropped down to 0.8 nmol L$^{-1}$
(Station 1 at 20 m depth). PFe concentrations increased with depth at all three stations and reached a maximum
at the bottom of station 2 (138.5 m) with more than 300 nmol L$^{-1}$ of PFe.  Lithogenic tracers, such as PAl or





PMn, presented similar profiles to PFe with concentrations ranging between 0.11 and 1544 nmol L$^{-1}$, and from
below detection to 2.51 nmol L$^{-1}$ respectively. The highest concentrations were also measured at the bottom of
station 2 (138.5 m). Total particulate phosphorus (PP) concentrations were relatively low in this area ranging
from undetectable values to 38 nmol L$^{-1}$. Maximum PP was measured in surface at Station 1 (20 m depth), then
concentrations decreased with depth and were less than 0.7 nmol L$^{-1}$ below 1000 m depth.


*3.4. Iberian Abyssal Plain (stations 11 to 17) and Western European Basin (stations 19 to 29)*

In the Iberian Abyssal Plain (IAP) and the Western European Basin (WEB), particulate iron concentration
vertical profiles were similar (Figure 4); median PFe concentrations were 0.18 nmol L$^{-1}$ in the first 100 m and
steadily increased with depth. Close to the seafloor, concentrations of PFe were up to 1.4 nmol L$^{-1}$ at every
station and reached values superior to 8 nmol L$^{-1}$ at stations 26 and 29, with low beam transmissometrytry
(<98%). Particulate aluminium profiles matched the PFe profiles, with low median concentrations within the
first 100m of 1.77 nmol L$^{-1}$ and 26 pmol L$^{-1}$ respectively. Then, concentrations increased with depth to reach a
maximum close to the oceanic floor. At stations 26 and 29, total PAl concentrations reached high values, up to
42 nmol L$^{-1}$. In the Western European Basin, PMn concentrations ranged from below detection to 0.36 nmol L$^{-1}$,
except close to the bottom of stations 26 and 29, where high concentrations of 0.91 and 1.31 nmol L$^{-1}$ were
measured, respectively. Particulate phosphorus profiles, while similar between stations of this basin, differed a
lot from the other element profiles. In the WEB, surface median PP concentration was two times higher than in
the Iberian margin (60 nmol L$^{-1}$ against 28 nmol L$^{-1}$ in the first 50 m with a maximum of 162 nmol L$^{-1}$ (station
21). Concentrations dropped drastically with depth and remained under 10 pmol L$^{-1}$ below 100 m.


*3.5. Icelandic Basin (stations 32 to 36)*

Concentrations of PFe were in a similar range and displayed analogous profiles to the ones collected in the
Western European Basin (figure 4), from below detection to 40.6 nmol L$^{-1}$; with low values at the surface (<1
nmol L$^{-1}$) and a progressive increase with depth. Close to the basin seafloor, low beam transmissometry (97.4%)
measurements were associated with high PFe concentrations of 40.6 nmol L$^{-1}$ at 3271 m of station 32.
Particulate aluminium vertical profiles were similar to those in the WEB but with extremely low surface
concentrations below 0.6 nmol L$^{-1}$; PAl then increased steadily with depth, reaching values up to 2 nmol L$^{-1}$
below 500 m. As previously observed for PFe, PAl concentrations were higher close to the seafloor, from 29
nmol L$^{-1}$ at station 34 to 101 nmol L$^{-1}$ at station 32. PMn also presented similar distributions than PFe and PAl.
Median surface concentrations were low within the first 100 m, 31 pmol L$^{-1}$ and 35 pmol L$^{-1}$, respectively, and
increased in the deep ocean to reach a maximum of 2.98 nmol L$^{-1}$ close to the seafloor. The Icelandic Basin had
a typical vertical profile for PP, with high concentrations at the surface, reaching 129 nmol L$^{-1}$ at station 32 and
really low concentrations below 150 m, inferior to 20 nmol L$^{-1}$.

*3.6. Reykjanes Ridge (station 38)*



Surface concentrations of particulate Fe, Al, and Mn above the Reykjanes Ridge (RR) were similar to the Icelandic Basin (Figure 3). However, close to the seafloor, high concentrations were measured, with PFe, PAl, and PMn reaching 16.2 nmol L$^{-1}$, 28.8 nmol L$^{-1}$, and 0.51 nmol L$^{-1}$ at 1354 m, respectively. Low concentrations of PP were measured in surface waters, with a median value of 24.8 nmol L$^{-1}$ in the top 100 m and a maximum of only 72.6 nmol L$^{-1}$ at 20 m.

### *3.7. Irminger Basin (stations 40 to 60; except Stations 53 and 56)*

Particulate Iron, Aluminium and Manganese distributions were similar to stations sampled in the WEB, IcB and IAP provinces (Figure 3). Surface concentrations of these elements were lower than 1.1 nmol L$^{-1}$, 3.4 nmol L$^{-1}$, and 0.4 nmol L$^{-1}$, respectively. Then, below 50 m depth, concentrations of PFe, PAl, and PMn increased and reached high values close to the seafloor, especially at stations 42 and 44; reaching up to 40 nmol L$^{-1}$, 90 nmol L$^{-1}$, and 1.5 nmol L$^{-1}$ respectively. Close to the Greenland margin, at the bottom of stations 49 and 60, concentrations of particulate trace metals were also elevated with PFe greater than 10 nmol L$^{-1}$. Particulate phosphorus concentrations were relatively high in surface waters, of the Irminger Basin, with a median value of 127 nmol L$^{-1}$ within the first 50 m. Particulate phosphorus decreased with depth and remained constant below 500 m with concentration below 10 nmol L$^{-1}$.

### *3.8. Greenland coast (stations 53, 56 and 61)*

Particulate Fe concentrations in the vicinity of the Greenland shelf had a high median concentration of 10.8 nmol L$^{-1}$, while PAl and PMn also had high median concentration of 32.3 nmol L$^{-1}$ and 0.44 nmol L$^{-1}$, respectively. Concentrations of PP were maximum at the surface with a value of 197 nmol L$^{-1}$ at 25 m of station 61. Then, PP concentrations decreased strongly with depth with values below 30 nmol L$^{-1}$ below 100 meters depth. Furthermore, beam transmissometry values in surface waters at these two stations, were the lowest of the entire section, with values below 85 %.

### *3.9. Labrador Basin (stations 63 to 77)*

In the Labrador Basin, median concentrations of PFe within the first 100 m were low, with a median value of 0.9 nmol L$^{-1}$ (n=30). However, at two stations, elevated concentrations were determined, up to 4.4 nmol L$^{-1}$ at station 77 at 40 m and 7 nmol L$^{-1}$ at station 63 between 70 and 100 m depth. Below the surface waters, PFe remained constant with depth until in proximity of the seafloor (Fig. 3). Between stations 64 and 71, the median concentration between 100 m and 200 m above the seafloor was 2.0 nmol L$^{-1}$ (n=39). Particulate Fe concentration at station 63, close to the Greenland margin, remained constant below 100 m depth, with a high median value of 5.7 nmol L$^{-1}$. On the other side of the Labrador Basin, station 77, close to the Newfoundland margin, constant PFe values of 3 nmol L$^{-1}$ between the surface and 200 m above bottom depth were observed. As previously described, PFe concentration increased close to the seafloor to 88 nmol L$^{-1}$ at station 71 at 3736 m. Particulate Al and Mn displayed similar characteristics to PFe, with low median concentrations at the surface of 3.37 nmol L$^{-1}$ and 90 pmol L$^{-1}$ respectively. Close to the seafloor of Station 71, at 3736 m depth, PAl, and





PMn reached high concentrations of 264 and 3.5 nmol L$^{-1}$. Particulate Phosphorus distribution was no different
than in the eastern basins, with 71 nmol L$^{-1}$ median PP within the first 50 m. Then below 50 m, the
concentration dropped off quickly to a median PP of 3 nmol L$^{-1}$.

*3.10. Newfoundland Shelf (station78)*
Close to the Newfoundland margin, surface waters displayed a small load of particulate trace metals as PFe,
PAl, and PMn were below 0.8 nmol L$^{-1}$, 2 nmol L$^{-1}$, and 0.15 nmol L$^{-1}$ respectively. Then close to the bottom, at
371 m, beam transmissometrytry values dropped to 94% and were associated with extremely high
concentrations of lithogenic elements: PFe=168 nmol L$^{-1}$, PAl=559 nmol L$^{-1}$, and PMn=2 nmol L$^{-1}$. Total PP
concentrations in the first 50 m ranged from 35 to 97 nmol L$^{-1}$. Below the surface, PP remained relatively high
with values up to 16 nmol L$^{-1}$ throughout the water column.

**4. Discussion**
Our goal in this work was to investigate mechanisms that drive the distribution of PFe in the North Atlantic, in
particular the different routes of supply and removal.
Possible candidate sources of PFe include lateral mixing from the different margins, atmospheric inputs,
recently melted sea ice, melting ice shelves and icebergs, resuspended sediments or hydrothermal inputs and
biological pool. Removal processes include remineralization and dissolution processes.
In the following sections, we examine each of these sources and processes, explore the evidence for their
relative importance, and use compositional data to estimate the particle types and host phases for iron and
associated elements.
*4.1. Analysis of the principal factors controlling variance: near-ubiquitous influence of*
*crustal particles in the water column*
Positive Matrix Factorisation (PMF) was run to characterise the main factors influencing the particulate trace
elements variances along the GEOVIDE section. In addition to PFe, PAl, PMn, and PP, nine additionnal
elements were included in the PMF: Y, Ba, Pb, Th, Ti, V, Co, Cu and Zn. The analysis has been conducted on
samples where all the 13 elements previously cited are above the detection limits; after selection, 445 of the 549
existing data points were used. Analyses were performed using the PMF software, EPA PMF 5.0, developed by
the USA Environmental Protection Agency (EPA). Models have been tested with several factors number (from
3 to 6), after full error estimation of each model, we decide to use the configuration providing the lowest errors
estimations and in consequence the most reliable.
In consequence, models were set up with four factors and were run 100 times to observe the stability of the
obtained results. After displacement, error estimations and bootstraps error estimations, the model was
recognised as stable. Results are shown in Figure 5.
The first factor is characterised by lithogenic elements, representing 86.8% of the variance of PFe, 75.8% of PAl
and 90.5% of PTi. The second factor is correlated with both Mn and Pb and explains no less than 76.5% and
77.0% of their respective variances. Ohnemus and Lam (2015) observed this co-relation between manganese



and lead particles and explained it by the co-transport on Mn-oxides (Boyle et al., 2005). The formation of barite
is causing the third factor constraining 87.7% of the Ba variance in the studied regions. A biogenic component is
the fourth factor and explained most of particulate phosphorus variance, 83.7%. The micronutrient trace metals,
copper, cobalt and zinc, had more than a quarter of their variances influenced by this factor.


Along the GA01 section, PFe distributions were predominantly controlled by lithogenic material, and to a
smaller extent by remineralisation processes (as seen by a Factor 3 contribution of 4.1%). These inputs and
processes are discussed below.
To further investigate the influence of crustal material on the distribution of PFe, it is instructive to examine the
distribution of the molar ratio of PFe/PAl along the section as a way to assess the lithogenic inputs (Lannuzel et
al., 2014; Ohnemus and Lam, 2015; Planquette et al., 2009) (Figure 6) along the section.

The PFe/PAl ratio can be used to estimate the proportion of lithogenic particles within the bulk particulate
material. A comparison with the Upper Continental Crust (UCC) ratio of Taylor and McLennan (1995), 0.21,
was used to calculate the lithogenic components of particles (PFe$_{litho}$) following Eq. (1):

$$\%\text{PFe litho} = 100 * \left(\frac{\text{PAl}}{\text{PFe}}\right) \text{sample} * \left(\frac{\text{PFe}}{\text{PAl}}\right) \text{UCC ratio} \qquad (1)$$

Then the non-lithogenic PFe is simply obtained using Eq. (2):

$$\%PFe\ non-litho = 100 * \%PFe\ litho \qquad (2)$$

Overall, the lithogenic contribution to PFe varies from 24% (station 60, 950 m) to 100% at stations located
within the Western European Basin. The most striking feature is the almost exclusive lithogenic nature of PFe
from stations 1 to 26 throughout the water column, except between 1000 and 3000 m at stations 21 to 26 (Figure
6 and 7). This feature could be linked to the fact that atmospheric inputs generally dominate the supply of PFe -
deposited from Saharan dust and transported via the Gulf Stream and North Atlantic Current to the WEB
(Shelley et al., 2017; Garcia-Ibanez et al. (2015)), even if low atmospheric fluxes were reported during our
cruise.

This feature at 1000 and 2500 m between stations 21 and 26 is likely be associated to the presence of the Sub-
Arctic Front, located between 49.5 and 51°N latitude and 23.5 and 22°W longitude (Zunino et al., 2017).
Indeed, this front which separates cold and fresh water of subpolar origin from warm and salty water of
subtropical origin was clearly identifiable at station 26 by the steep gradient of the isotherms and isohalines. The
fact that the WEB was sampled close to but just after the bloom maximum is limiting any higher PFe/PAl
signatures (see also section 3.3.4). The intrusion of an old LSW at stations 21 to 26 between 1000 and 2500 m





with a different PFe/PAl signature could explain the smaller contribution of lithogenic PFe in this depth range as
atmospheric inputs to the Labrador Sea region are relatively small  (Shelley et al., 2017).

### 4.2. Fingerpriting watermasses

The GEOVIDE section crossed several distinct water masses along the North Atlantic, each of them being
distinguishable by their salinity and potential temperature signatures (García-Ibáñez et al., 2015; Figure 2).
Based on this study, we applied a Kruskal-Wallis test on molar PFe/PAl ratios of nine water masses (Figure 8)
in order to test the presence of significant differences. Water masses for which we had less than 5 data points for
PFe/PAl were excluded from this test.
As previously seen, the lithogenic inprint is dominant in the WEB, with MW and NEADW showing PFe/PAl
values close to the UCC value of 0.21 mol mol$^{-1}$. Interestingly, the PFe/PAl signature of 0.36 mol mol$^{-1}$ within
the old LSW$_{WEB}$ is probably due to the effect of biologic inputs associated with the strong bloom encountered in
the Irminger Sea than in the WEB (see section 4.3.5).
While it appears that lithogenic particles are dominating the water column in the WEB, and that some water
masses have a clear PFe/PAl fingerprint, it is important to discuss the origin of these signatures, which is the
purpose of the following sections.

### 4.3. Tracking the different inputs of particulate iron

4.3.1. Inputs at margins: Iberian, Greenland and Newfoundland

Inputs from continental shelves and margins have been demonstrated to support high productivity in shallow
coastal areas. Inputs of iron from continental margin sediments supporting the high productivity found in
shallow coastal regions have been demonstrated in the past (e.g. Cullen et al. (2009), Elrod et al. (2004), Jeandel
et al. (2011), Ussher et al. (2007)) and sometimes, were shown to be advected at great distances from the coast
(e.g. Lam et al., 2008).  Moreover, freshwater inputs that are usually present in these regions can also play a key
role in the global biogeochemical cycling of trace metals (Blain and Tagliabue, 2016; Guieu et al., 1991; Martin
and Meybeck, 1979). Rivers, runoff and continental glacial  melt and/or sea-ice melt can also supply dissolved
and particulate iron to coastal waters, thus sustaining important phytoplankton production (Fung, 2000).
In the following section, we will investigate these possible candidate sources in proximity of the different
margins encountered. Along the GEOVIDE section, sediments at margins were of various compositions
(Dutkiewicz et al., 2015). Sediments originating from the Iberian margin were mainly constituted of silts and
clays (Cacador et al., 1996; Duarte et al., 2014). East Greenland margin sediments were a mixture of sands and
grey/green muds, while, sediments from the West Greenland margin were mainly composed of grey/green muds
(Loring and Asmund, 1996). At the western end of the section, sediments from the Newfoundland margin were
composed of gravelly and sandy muds (Mudie et al., 1984). The different sediment compositions of the three
margins sampled during GEOVIDE have different mineralogy, which are reflected in their different PFe/PAl





ratios (Figure 9). While the Iberian Margin had a PFe/PAl close to UCC ratio, mainly due to seasonal dust inputs from North Africa,(Shelley et al., 2017) the highest biogenic contribution could be seen at the East Greenland (stations 53 and 56) and West Greenland (station 61) Margins, with median PFe/PAl reaching 0.45 mol mol$^{-1}$. The Newfoundland margin displayed an intermediate behaviour, with Fe/Al ratios of 0.35 mol mol$^{-1}$.

In addition to PAl, PMn can be used as a tracer of inputs from shelf resuspension (Lam and Bishop, 2008). Indeed, Mn is really sensitive to oxidation mediated by bacteria (Tebo et al., 1984; Tebo and Emerson, 1985) and forms manganese oxides ($MnO_2$). These authigenic particles lead to an enrichment of Mn in particle compositions. In order to track the influence of shelf resuspension, a percentage of sedimentary inputs "%bulk sediment inputs" can be calculated using PMn/PAl ratio from GEOVIDE samples and the PMn/PAl UCC value (0.0034; Taylor and McLennan, 1995) according to the following equation:

$$\%\text{bulk sediment PMn} = 100 * \left(\frac{PAl}{PMn}\right) \text{sample} * \left(\frac{PMn}{PAl}\right) \text{UCC ratio} \qquad (3)$$

This proxy is a good indicator of direct and recent sediment resuspension. We assume that particles newly resuspended in water column will have the same PMn/PAl ratio than the UCC ratio leading to a "%bulk sediment Mn" of 100%. This value will decrease by authigenic formation of Mn oxides. When a sample presents a "%bulk sediment Mn" greater than 100%, we assign a value of 100% to simplify the following discussion. As the Mn cycle can also be affected by biologic uptake (e.g. Peers and Price, 2004; Sunda and Huntsman, 1983), this proxy is only used at depths where biologic activity is negligible (i.e. below 150m depth).

*The Iberian margin*

Coastal waters of the Iberian Shelf are impacted by the runoff for the Tagus River, which is characterised by high suspended matter discharges, ranging between 0.4 to $1 \times 10^6$ tons yr$^{-1}$, and with a high anthropogenic signature (Jouanneau et al., 1998). During the GEOVIDE section, the freshwater input was observable at stations 1, 2 and 4 in the first 20 m; salinity was below 35.2 psu while surrounding waters masses had salinity up to 35.7 psu. Within the freshwater plume, particulate concentrations were important at station 2, at 20m, PFe was 1.83 nmol L$^{-1}$. Further away from the coast, the particulate concentrations remained low at 20m depth, with PFe, PAl, and PMn concentrations of 0.77 nmol L$^{-1}$, 3.5 nmol L$^{-1}$, and 0.04 nmol L$^{-1}$, respectively at station 1. The low expansion of the Tagus plume is likely due to the rapid settling of suspended matter. Indeed, our coastal station 2 was already located at around 50 km of the Iberian coast and according to Jouanneau et al. (1998), the surface particle load can be observable at a maximum 30km of the Tagus estuary.

Besides, ADCP data acquired during GEOVIDE (Zunino et al., 2017) and several studies have reported an intense current spreading northward coming from Strait of Gibraltar and Mediterranean Sea, leading to a strong resuspension of benthic sediments above the Iberian Shelf e.g. Biscaye and Eittreim (1977), Eittreim et al. (1976), McCave and Hall (2002), Spinrad et al. (1983). The importance of the sediment resuspension was observable by low beam transmissometry value (87.6%) at the bottom of station 2. This important sediment resuspension led to an extensive input of lithogenic particles within the water column associated with high concentrations of PFe (304 nmol L$^{-1}$), PAl (1500 nmol L$^{-1}$), and PMn (2.5 nmol L$^{-1}$) (Figure 3, Table S1).





Moreover, one hundred percent of PFe is estimated to have a lithogenic origin (Figure 11) while 100% of the PMn was the result of a recent sediment resuspension according to the %Fe$_{litho}$ and "%bulk sediment Mn" proxies (Table S1), confirming the resuspended particle input.

At distance from the shelf, within the Iberian Abyssal Plain, an important lateral advection of PFe from the margin was observable (Figure 11). These lateral inputs occurred at two depth ranges: between 400 and 1000 m at seen at stations 4 and 1, with PFe concentrations reaching 4 nmol L$^{-1}$, and between 2500 m and the bottom (3575 m) of station 1, with PFe concentrations reaching 3.5 nmol L$^{-1}$. While 100% of PFe had a lithogenic origin, the sedimentary source input decreased, between 40% and 85% of the PMn (Figure 11). Transport of lithogenic particles was observable until station 11 (12.2°W) at 2500 m where PFe concentration was 7.74 nmol L$^{-1}$ and 60% of PMn had a sedimentary origin (Figure 10). Noteworthy, no particular increase in PFe, PMn or PAl was seen between 500 and 2000 m depth, where the MOW spreads, which is consistent with that was observed DFe concentrations (Tonnard et al., this issue), yet in contrast with the dissolved aluminium values (Menzel Barraqueta et al, subm., this issue) which were high in the MOW and with the study of Ohnemus and Lam (2015) that reported a maximum PFe concentration at 695 m depth associated with the particle-rich Mediterranean Overflow Water (Eittreim et al., 1976) in the IAP. However, their station was located further south of our station 1. The shallower inputs observed at stations 1 and 4 could therefore be attributed to sediment resuspension from the Iberian margin and nepheloid layer at depth for station 1.

Therefore, the Iberian margin appears to be an important source of lithogenic-derived iron-rich particles in the Atlantic Ocean; shelf resuspension impact was perceptible until 280 km away from the margin (Station 11) in the Iberian Abyssal Plain.

*South Greenland*

Several studies already demonstrated the importance of icebergs and sea ice as source of dissolved and particulate iron (e.g. van der Merwe et al., 2011a, 2011b; Planquette et al., 2011; Raiswell et al., 2008). The Greenland shelf is highly affected by external fresh water inputs as ice-melting or riverine runoff (Fragoso et al., 2016), that are important sources of iron to the Greenland Shelf (Bhatia et al., 2013; Hawkings et al., 2014; Statham et al., 2008).

Both East and West Greenland shelves (stations 53 and 60) had high concentration of particles (beam transmissometry of 83%) and particulate trace elements, reaching 22.1 nmol L$^{-1}$ and 18.7 nmol L$^{-1}$ of PFe, respectively (station 53 at 100m and station 61 at 136 m). During the cruise, the relative freshwater observed (S<33 psu) within the first 25 meters of stations 53 and 61 were associated with high PFe (19 nmol L$^{-1}$), PAl (61 nmol L$^{-1}$), PMn (0.6 nmol L$^{-1}$) and a low beam transmissometry (≤ 85%) (Figure 10 and Table S1). Particles associated were enriched in iron compared to aluminium, as PFe/PAl ratio was 0.3 within the meteoric water plume. High biological production, in agreement with PP concentrations reaching 197 nmol L$^{-1}$ induced by the supply of bioavailable dissolved iron from meteoric water (Raiswell et al., 2008; Statham et al., 2008;, Tonnard et al., submitted, this issue), leaded to a transfer of DFe to the particulate phase. This is in line with the fact that around 30% of the PFe had a non-lithogenic and likely biogenic origin. In addition, only 35% of the PMn originated from resuspended sediments. Interestingly, these two proxies remained constant from the seafloor to the surface (Station 49, Figure 11), with around 25% of the PMn of sedimentary origin, which could be due to



an important mixing happening on the shelf. The lithogenic PFe could result from the release of PFe from
Greenland bedrock captured during the ice sheet formation on land.
The spatial extent of the off-shelf lateral transport of particles was not important on the east Greenland coast.
Indeed, no visible increase of particulate trace metal concentrations was visible at the first station off-shelf,
station 60 (Figure 11), except at 1000 m depth, where a strong increase (up to 89%) of sedimentary PMn was
seen. This is probably due to the East Greenland Coastal Current (EGCC) that was located at station 53
constrained these inputs while stations 56 and 60 were under the influence of another strong current, the East
Greenland-Irminger current (EGIC) (Zunino et al., 2017).

To the west of the Greenland margin, lateral transport of particles was slightly more important. Noticeable
concentrations of particulate lithogenic elements were observable until station 64 located 125 km away from
shoreline. These particles had decreasing PFe lithogenic contribution (52%) with a similar (27%) sedimentary
PMn content than closer to the margin. The increasing nature of non-lithogenic PFe is linked to the bloom in
surface (associated with a PFe/PAl ratio of 0.30 mol mol$^{-1}$ and a PP of 197 nmol L$^{-1}$ at station 61), with the
biogenic PFe settling down along the transport of particles.

Therefore, particles newly resuspended from Greenland sediments are an important source, representing around
a third of the pMn pool, combined with surface inputs such as riverine runoff and/or ice-melting that are
delivering particles on the shelf and biological production. Unlike the Iberian shelf, Greenland margin was not
an important provider of particulate metals inside the Irminger and Labrador Basin, due to the circulation that
constrained the extent of the margin plume.

*The Newfoundland Shelf*
Previous studies already described the influence of fresh water on the Newfoundland shelf from the Hudson
Strait and/or Canadian Artic Archipelago (Fragoso et al., 2016; Yashayaev, 2007). Yashayaev (2007) also
monitored strong resuspension of sediments associated with the spreading of Labrador Current along the West
Labrador margin.
Close to the Newfoundland coastline, at station 78, high fresh water discharge (≤ 32 psu) was observed in
surface (Benetti et al., 2017). Interestingly, these freshwater signatures were not associated with elevated
particulate trace metal concentrations. Distance of meteoric water sources implied a long travel time for the
water to spread through the Labrador Basin to our sampling stations. Along the journey, particles present
originally may have been removed from water column by gravitational settling.
The proportion of lithogenic PFe was relatively high and constant in the entire water column, with a median
value of 67 %. At station 78, 95% of the PMn had a sedimentary origin close to the seafloor (371 m). The
spreading of the recent sediment resuspension was observable until 140 m depth where the contribution of
sedimentary Mn was still 51% (Figure 11, Table S2). This could correspond to an intense nepheloid layer as
previously reported by Biscaye and Eittreim (1977) (see also section 3.3.2). The high PFe concentration (184
nmol L$^{-1}$, station 78, 371 m) associated with a high percentage of sedimentary PMn (95%) observed at the
bottom of this station, was therefore the result of an important resuspension of shelf sediments. This was
confirmed with low transmissometry values of 95%.





Despite the important phytoplanktonic community present (Tonnard et al., in prep), the PFe remained low at
0.79 nmol L$^{-1}$ at 10 m, but, the high PFe/PAl ratio, up to 0.4, and the PP concentration of 97 nmol L$^{-1}$, confirms
the biologic influence. Either the biogenic particles settled quickly, and/or they were quickly remineralized.
Concerning this latter process, intense remineralization at station 77 has been reported by Lemaitre et al.
(2018a), which could explain the low PFe values throughout the water column. That said, resuspended particles
are were still laterally transported off-shelf until station 71 (Figure 6) where PFe concentrations were higher
than the background value, up to 2 nmol L$^{-1}$ at depths greater than 100 m.

Along the GEOVIDE section, continental shelves provided an important load of particles within the surrounding
water column. The three margins sampled during GEOVIDE behaved differently; the Iberian margin discharged
high quantities of lithogenic particles far away from the coast while the Greenland and Newfoundland margins
did not reveal important PFe concentrations. Spreading of particles is tightly linked to hydrodynamic conditions,
which in the case of the Greenland margin, prevented long distance seeding of PFe. Moreover, each margin
showed a specific PFe/PAl ratio (Figure 9) indicating different composition of the resuspended particles.
Resuspended particles represent the composition of sediment at the margin if oxido-reductive transformation of
iron and aluminium are considered negligible under these circumstances. Differences between margins were due
to the presence of non-crustal particles. Biological production in surface waters produces particles with a higher
PFe/PAl content and their export through the water column to the sediment increased the PFe/PAl ratio at depth.
Regions where biological production is intense such as in the vicinity of Newfoundland presented higher
PFe/PAl ratios of resuspended benthic particles. These results are in agreement with the study of Lam et al.
(2017), which showed the different behaviour between margins are a function of several parameters such as
boundary currents, internal waves and margin sediment composition.

4.3.2 Benthic resuspended sediments

Benthic nepheloid layers (BNLs) can play a significative role in trace element distributions at depth as
previously described (Dutay et al., 2015; Lam et al., 2015; Ohnemus et al., 2015; Revels et al., 2015). BNLs are
important layers where local resuspension of sedimentary particles (Bishop and Biscaye, 1982; Eittreim et al.,
1976; Rutgers Van Der Loeff et al., 2002) occur due to strong hydrographic stresses interacting with the ocean
floor (Gardner et al., 2017). In the North Atlantic, boundary currents were suspected to be the origin of theses
stresses (Biscaye and Eittreim, 1977; Eittreim et al., 1976) but more recent studies demonstrate the essential role
of benthic storms and deep eddies (Gardner et al., 2018). Along the GA01 section, BNLs were observable in
each province with different strengths (Figures 3 and 12).

In BNLs located within the WEB, PFe concentrations reached up to 10 nmol L$^{-1}$ (stations 26 and 29). These
concentrations were smaller than PFe concentrations encountered in BNL from the Icelandic, Irminger and
Labrador Basins, where benthic resuspension led to PFe concentrations higher than 40 nmol L$^{-1}$, even reaching
89 nmol L$^{-1}$ at the bottom of station 71 (3736 m). Moreover, in the Irminger and Labrador Basins, PFe/PAl
molar ratios within BNLs were higher than the ones measured within the WEB at station 26 and 29. In the
Irminger Basin, PFe/PAl reached 0.4 mol mol$^{-1}$, which could reveal a mixture of lithogenic and biogenic matter




previously exported. This feature was also observed in the Labrador Basin, with PFe/PAl ratio ranging between
0.34 and 0.44 mol mol$^{-1}$. In contrast, BNLs sampled in the WEB have clearly a lithogenic imprint, with PFe/PAl
molar ratios close to the crustal one. Resuspended sediments with a non-crustal contribution seem to hold a
higher PFe content than sediments with a lithogenic characteristic. Nevertheless, interestingly all BNLs present
during GEOVIDE were spreading identically, with impacts observable up to 200 meters above the oceanic
seafloor (Figure 12), as reflected in beam transmissometry values, and PFe concentrations, that returned to a
background level at 200 m above the seafloor. The presence of these BNLs has also been reported by Le Roy et
al. (submitted, this issue).
Important differences of PFe intensities could also be due to different hydrographic components and
topographic characteristics. As previously explained, two main triggers of BNLs are benthic storms and deep
eddies; by definition these processes are highly variable geographically and temporally, but no physical data
could allow us to investigate further this hypothesis.

4.3.3. Reykjanes Ridge inputs

Recently, hydrothermal inputs of iron in the open ocean have been re-evaluated by (Fitzsimmons et al., 2017;
Resing et al., 2015; Tagliabue et al., 2014). These studies demonstrated the importance of hydrothermal
activities on the global iron biogeochemical cycle through particulate and dissolved iron fluxes. During the
cruise, samples of station 38 have been collected above the Reykjanes Ridge, the upper section of the Mid-
Atlantic Ridge in the North Atlantic, which has inferred hydrothermal sites from several studies conducted in
the area (Baker and German, 2004). Above the ridge, high PFe concentrations were measured, reaching 16 nmol
L$^{-1}$ just above the seafloor, while increased DFe concentrations were reported to the East of the ridge (Tonnard
et al., this issue).. The exact sources of iron-rich particles cannot be well constrained, as they could come from
active hydrothermal vents or resuspension of particulate matter from new crustal matter produced at the ridge.
According to the oceanic circulation (Zunino et al., 2017; Garcia-Ibanez et al., 2017), hydrothermal particles
could have been seen in the ISOW within the Icelandic Basin. Nevertheless, at the vicinity of the ridge, scanning
electron microscope (SEM) analyses of our samples did reveal a number of biological debris and clays but not
the presence of iron (oxy-)hydroxide particles, which are known to be highly produced close to hydrothermal
vents (Elderfield and Schultz, 1996). Their absence could thus indicate an absence of vents. However, other
proxies, such as helium-3, are necessary to claim with more accuracy the presence or absence of an
hydrothermal source close to station 38.
Alternatively, resuspended sediments transported with ISOW flowing across the Reykjanes Ridge could explain
the high PFe concentrations below 1000 m depth at station 38. This feature was associated with lower median
PFe concentrations and PFe/PAl ratios (Figure 7) at station 40 (2.2 nmol L$^{-1}$, and 0.58 mol mol$^{-1}$ respectively)
than at station 38 (6.8 nmol L$^{-1}$, 0.48 mol mol$^{-1}$ respectively). Moreover, PMn had a 19% sedimentary origin,
constant from the bottom to 1163 m depth, a contribution that is very low for the shallower water depths.
Consequently, the increase in PFe within the ISOW$_{west}$ more likely came from sediment resuspension as the
ISOW$_{east}$ flows through the Charlie Gibbs Bight Fracture Zones.

4.3.4. Atmospheric inputs





Atmospheric deposition is an important input of trace elements in surface of the open ocean (e.g. (Jickells et al.,
2005). Atmospheric inputs, both wet and dry, were reported to be low during the GEOVIDE cruise (Menzel-
barraqueta et al., 2018, this issue; Shelley et al., 2017; 2018). In fact, oceanic particles measurements in surface
waters along the section did not reveal high PFe or PAl concentrations, therefore, the surface composition of
particles did not seem to be highly affected by atmospheric deposition at the time of the cruise. However,
PFe/PAl ratio was closed to the UCC one, probably due to the overall influence of atmospheric deposition in
this area. One pattern is also interesting to note: the surface waters of the Iberian Abyssal Plain and Western
European Basin, between stations 11 and 23 presented a characteristic feature with really low PFe/PAl
elemental ratios, of 0.11, smaller than the UCC ratio of 0.21 (Figure 6). Such low ratios have been reported in
the same region by Barrett et al. (2012). One possible explanation is given by Buck et al. (2010) who described
Fe-depleted aerosols in this area of the North Atlantic with PFe/PAl ratio below UCC ratio. However, Shelley et
al. (2017) found a higher PFe/PAl ratio around 0.25 is this area (their sample geoa5-6). This result, highlights
some of the difficulties that link atmospheric inputs to water column data (Baker et al., 2016), and implies a
probable fractionation after aerosol deposition. In addition, there is high spatial and temporal variability of
atmospheric deposition (Mahowald et al., 2005) and a certain degree of uncertainty about the dissolution
processes of atmospherically-transported particles (Bonnet and Guieu, 2004).

4.3.5. Influence of phytoplankton assemblages, remineralisation and scavenging in

the upper water column

Biological activity in surface waters impacts the particle composition in the upper water column. In bulk particle
samples, direct measurement of the biogenic metal fraction is not possible due to the heterogeneity of particles,
and in particular, the presence of lithogenic particles. It is however possible to estimate the $PFe_{nonlitho}$/PP molar
ratios, based on Eq. (1) and (2), and assuming that most of the PP is of biogenic origin. As 100% of the PFe was
estimated to be of lithogenic origin, stations 1 to 26 are excluded from the discussion below.
Overall, the median $PFe_{nonlitho}$/PP molar ratios varied from 1.0 (Irminger Basin) to 38.7 mmol mol$^{-1}$ (Greenland
margin) in the upper 50 m. These ratios are consistent with the few available bulk PFe/PP ratios available in the
literature (Twining and Baines, 2013 and references therein), ranging from 1 to 31 mmol mol$^{-1}$ and the
phytoplankton assemblages encountered during GEOVIDE (Tonnard et al., in prep.). Indeed, the highest
$PFe_{nonlitho}$/PP molar ratio determined at stations 53 and 56 close to the South Greenland margin coincide with a
bloom mostly composed of large diatoms, whereas, the smallest ratios were associated with a bloom mainly
composed of cyanobacteria and haptophytes. The effect of biological uptake is also clearly visible when looking
at PFe/PAl vertical variation, which increases from the surface to approximately 100m depth (Figure 13), except
in the Iberian Margin, which is under the strong influence of lithogenic inputs.


At deeper depths, pelagic remineralisation processes influence the composition of particles (Barbeau et al.,
1996, 2001; Boyd et al., 2010; Strzepek et al., 2005).
Close to the IM and within the IAP, no PFe/PAl decrease that could point to a preferential remineralisation of
PFe over PAl could be observed within the remineralisation depth (200 to 400 m depth, Figure 13), whereas



preferential remineralisation of PP over PFe occurs, as reflected by increasing PFe/PP ratios (Figure 14). This is
probably due to the fact that remineralisation rates were low (Lemaitre et al., 2018a), and that PFe was mostly of
lithogenic origin, more difficult and slow to remineralize (Boyd et al., 2010). Below 600 m depth, scavenging
processes could explain the increasing PFe:PP ratios, from 0.30 to 0.80 mol mol$^{-1}$ at station 13.
Within the WEB, between 200 and 500 m depth, remineralisation of PFe over PAl occurs, although reported to
be small (Lemaitre et al., 2018a) as reflected by decreasing PFe:PAl ratios (Figure 13), while PFe:PP ratios
remained constant, pointing out to similar remineralisation rates of PFe and PP. Below 600 m depth, a stronger
scavenging of DFe onto particles than in IM and IAP is likely to explain the increasing ratios of PFe:PAl from
0.18 to 0.30 mol mol$^{-1}$ and PFe:PP from 0.047 to 0.367 (Station 21), and from 0.16 to 1.05 (Station 26) mol mol$^{-}$
$^{1}$. Similar patterns occur in IcB (station 32).

Above the RR, at station 38, PFe is remineralized preferentially over PAl and PP, with decreasing PFe:PAl
ratios from 0.46 to 0.19 mol mol$^{-1}$ and decreasing PFe:PP ratios from 0.24 to 0.04 mol mol$^{-1}$.  This interesting
feature is associated with moderate POC remineralisation fluxes (Lemaitre et al., 2018a), and the fact that a
stronger fraction of PFe was associated with biogenic material, easier to recycle.
In the IrB, PP is preferentially remineralized over PFe and PAl, as reflected by increasing PFe:PP ratios and
constant PFe:PAl ratios within the remineralisation depth. This is associated with high POC remineralisation
fluxes (Lemaitre et al., 2018a) and a high proportion of lithogenic PFe.
Finally, within the LB, PFe:PAl and PFe:PP remain constant within the deep remineralisation depth, extending
from 200 to 1000 m depth due to the deep convection of the LSW (Lemaitre et al., 2018a). Below 1000 m,
PFe:PP ratios increase from 0.29 to 0.85 mol mol$^{-1}$, while PFe:PAl ratios still remain constant. This could be
explained by the fact that most PP has been recycled due to the strongest remineralisation fluxes reported in this
area (Lemaitre et al., 2018a).

**5. Conclusions**
This investigation of the PFe compositions of suspended particulate matter in the North Atlantic indicates the
pervasive influence of crustal particles, augmented by sedimentary inputs at margins, and at depths, within
benthic nepehloid layers.
Indeed, along the GEOVIDE section, continental shelves provided an important load of particles within the
surrounding water column, with PFe mostly residing in non-biogenic particulate form. The Iberian margin
discharged high quantities of lithogenic particles originating from riverine inputs far away from the coast while
the Greenland margin did not reveal a long distance seeding of PFe, due to hydrodynamic conditions. Both
Greenland and Newfoundland margins PFe resuspended particles were under a strong biogenic influence that
were exported at depth. This resulted in different remineralisation fluxes among the different provinces.
Scavenging processes could also be visible at depths greater than 1000 m, these effects being the most
pronounced within the WEB.
Finally, resuspended sediments above the Reykjanes Ridge increased the PFe composition of the Iceland





Scottish Overflow Water. A similar feature occurs for the Labrador Sea Water, as it flows from the Irminger
Basin to the Western European Basin.



**Acknowledgments**
We are greatly indebted to the captain and crew of the N/O Pourquoi Pas? for their help during the GEOVIDE
mission and clean rosette deployment. We would like to give special thanks to Fabien Pérault and Emmanuel de
Saint Léger for their technical expertise, to Catherine Schmechtig for the GEOVIDE database management and
Greg Cutter for his guidance in setting up the new French clean sampling system. We also would like to thanks
Reiner Schlitzer for the Ocean Data View software (ODV).
This work was supported by the French National Research Agency (ANR-13-BS06-0014, ANR-12-PDOC-
0025-01), the French National Centre for Scientific Research (CNRS-LEFE-CYBER), the LabexMER (ANR-
10-LABX-19), and Ifremer. It was supported for the logistic by DT-INSU and GENAVIR.

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



**Figure 1: Map of stations where suspended particle samples were collected with GO-FLO bottles during the GEOVIDE cruise (GA01). Biogeochemical provinces are indicated by red squares, IM: Iberian Margin, IAP: Iberian Abyssal Plain, WEB: Western European Basin, IcB: Iceland Basin, RR: Reykjanes Ridge, IrB: Irminger Basin, GS: Greenland Shelf, LB: Labrador Basin, NS: Newfoundland Shelf. This figure was generated by Ocean Data View (Schlitzer, R., Ocean Data View, odv.awi.de, 2017).**

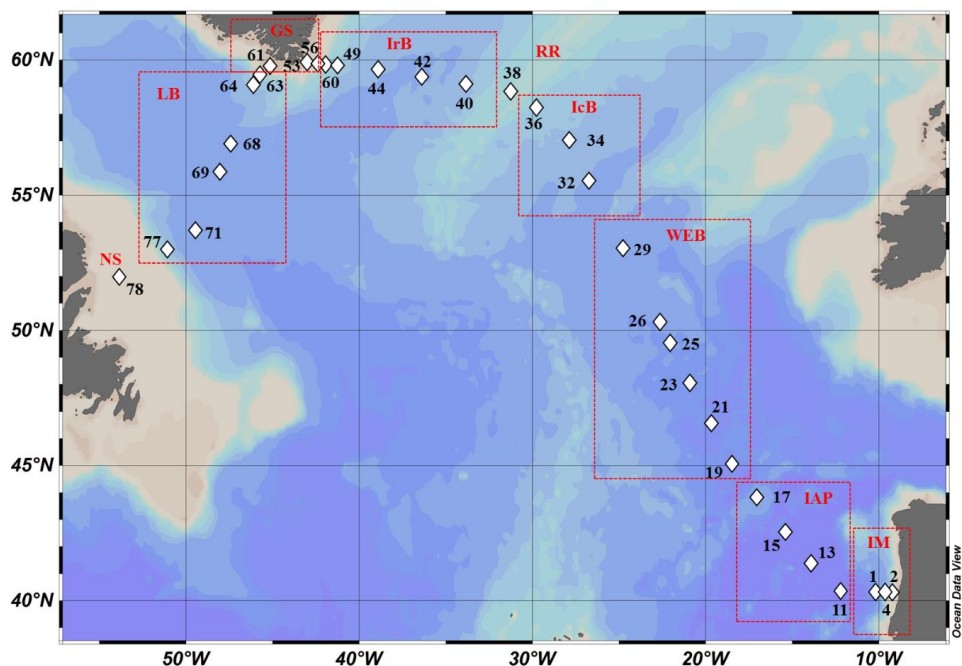



**Figure 2: Salinity section during the GEOVIDE cruise. Water masses are indicated in black, MW: Mediterranean Water; NACW: North Atlantic Central Water; NEADW: North East Atlantic Deep Water; LSW: Labrador Sea Water; ISOW: Iceland-Scotland Overflow Water; SAIW: Sub-Arctic Intermediate Water; IcSPMW: Iceland Sub-Polar Mode Water; IrSPMW: Irminger Sub-Polar Mode Water. Stations locations are indicated by the numbers. Biogeochemical provinces are indicated in blue font above station numbers. Contour of salinity = 35.8psu have been apply to identify the Mediterranean Water. This figure was generated by Ocean Data View (Schlitzer, R., Ocean Data View, odv.awi.de, 2017).**

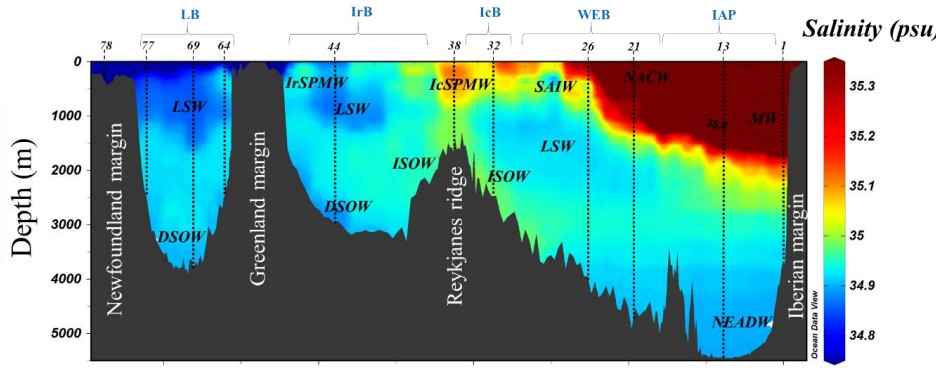





**Figure 3: Left) Distribution of total particulate iron (a, PFe), aluminium (b, PAl), manganese (c, PMn) and**
**phosphorus (d, PP) concentrations (in nmol L$^{-1}$) along the GEOVIDE section. Right) Contribution of small size**
**fraction (0,45-5 μm) expressed as percentage (%) of the total concentration of PFe (e), PAl (f), PMn (g) and PP (h).**
**Station IDs and biogeochemical region are indicated on top of section a. This figure was generated by Ocean Data**
**View (Schlitzer, R., Ocean Data View, odv.awi.de, 2017).**

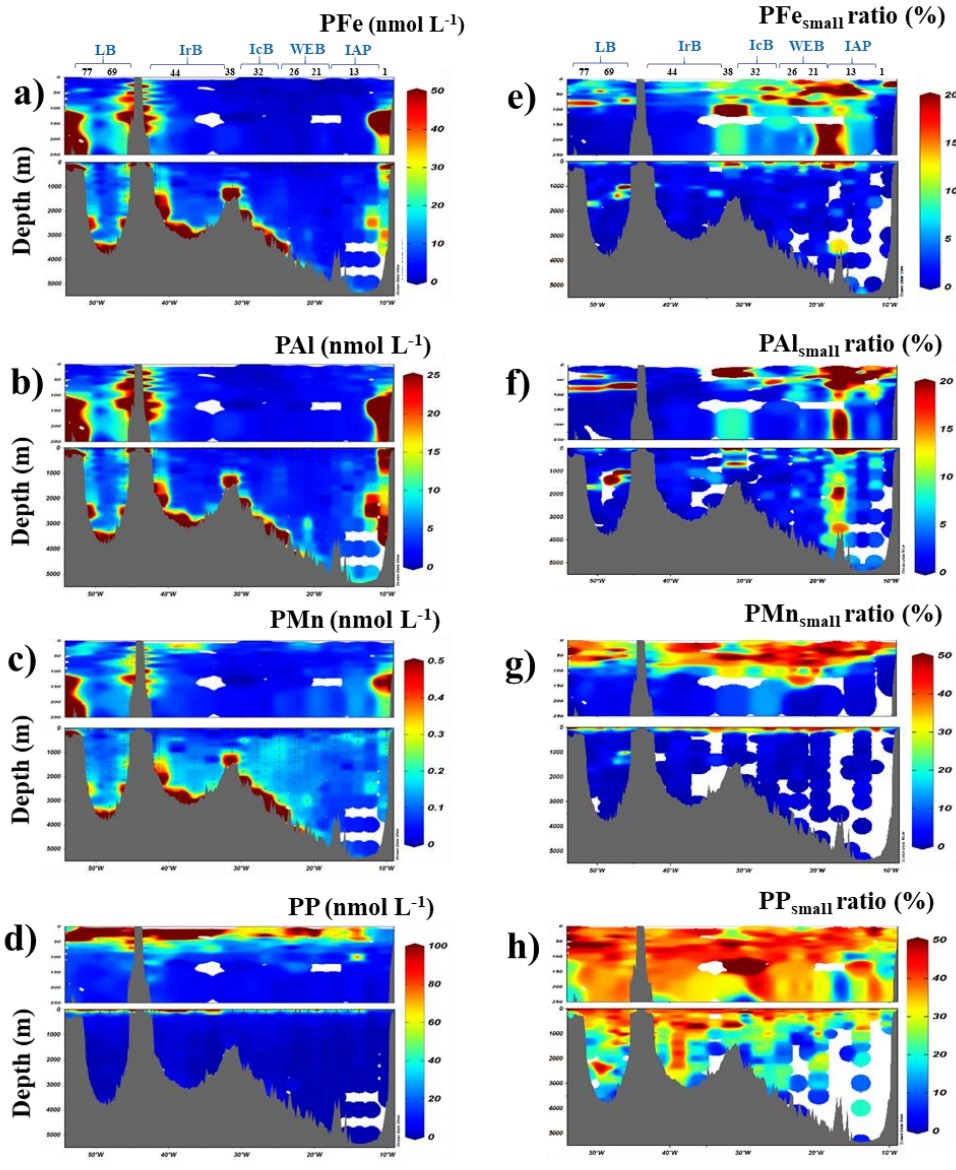









**Figure 4: Boxplot figure of the particulate iron vertical profile (in nmol L$^{-1}$) in the a) Iberian abyssal plain (IAP), b) Western European basin (WEB), c) Icelandic basin (IcB), d) Irminger basin (IrB) and e) Labrador basins (LB). The left boundary of the box represents the 25$^{th}$ percentile while the right boundary represents the 75$^{th}$ percentile, the line within the box marks the median value. Whiskers represent the 90$^{th}$ and 10$^{th}$ percentiles and dots are the outlying data. Seven depth boxes have been used (0-100m, 100-200m, 200-500m, 500-1000m, 1000-2000m, 2000-3000m and 3000m-bottom depth).**

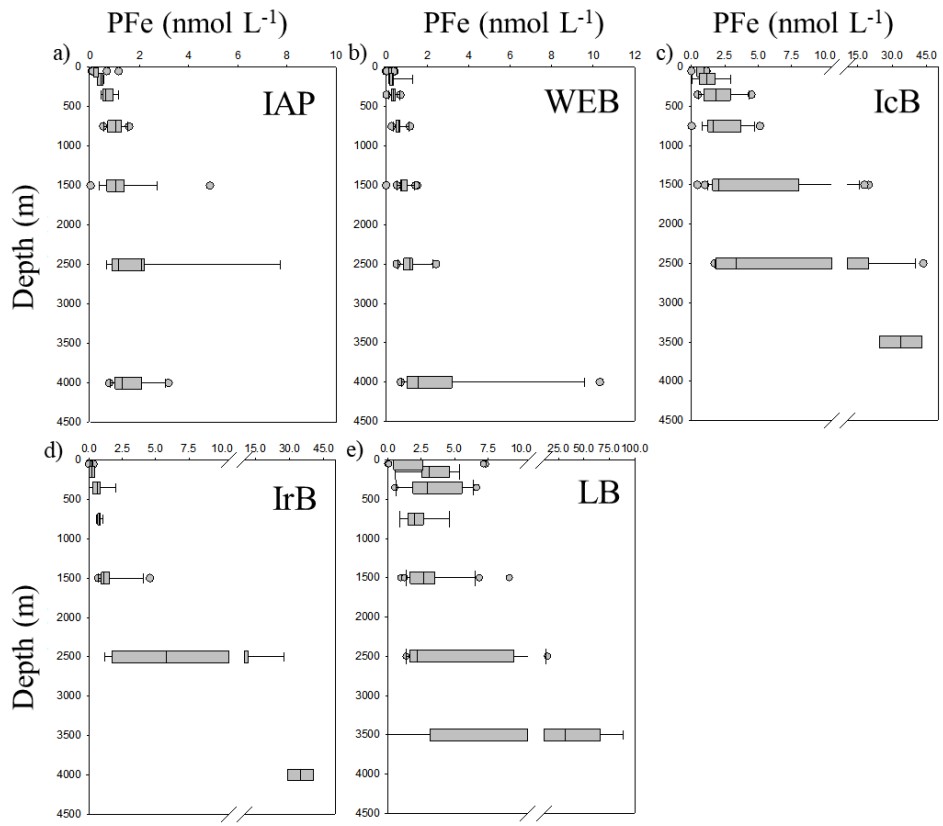



**Figure 5: Factor fingerprint of the positive matrix factorisation. The four factors are represented in a stacked bar**
**chart of the percentage of variance explained per element.**

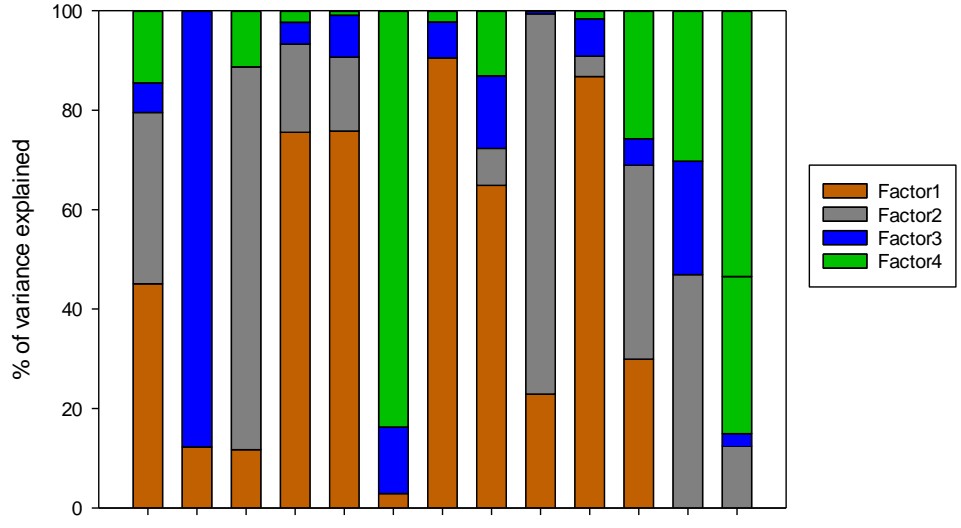






















**Figure 6: a) Section of the PFe to PAl molar ratio (mol mol$^{-1}$); (b) contribution of lithogenic PFe (%) based on Eq. (1).**
**Station IDs and biogeochemical provinces are indicated above each section. This figure was generated by Ocean Data**
**View (Schlitzer, R., Ocean Data View, odv.awi.de, 2017).**

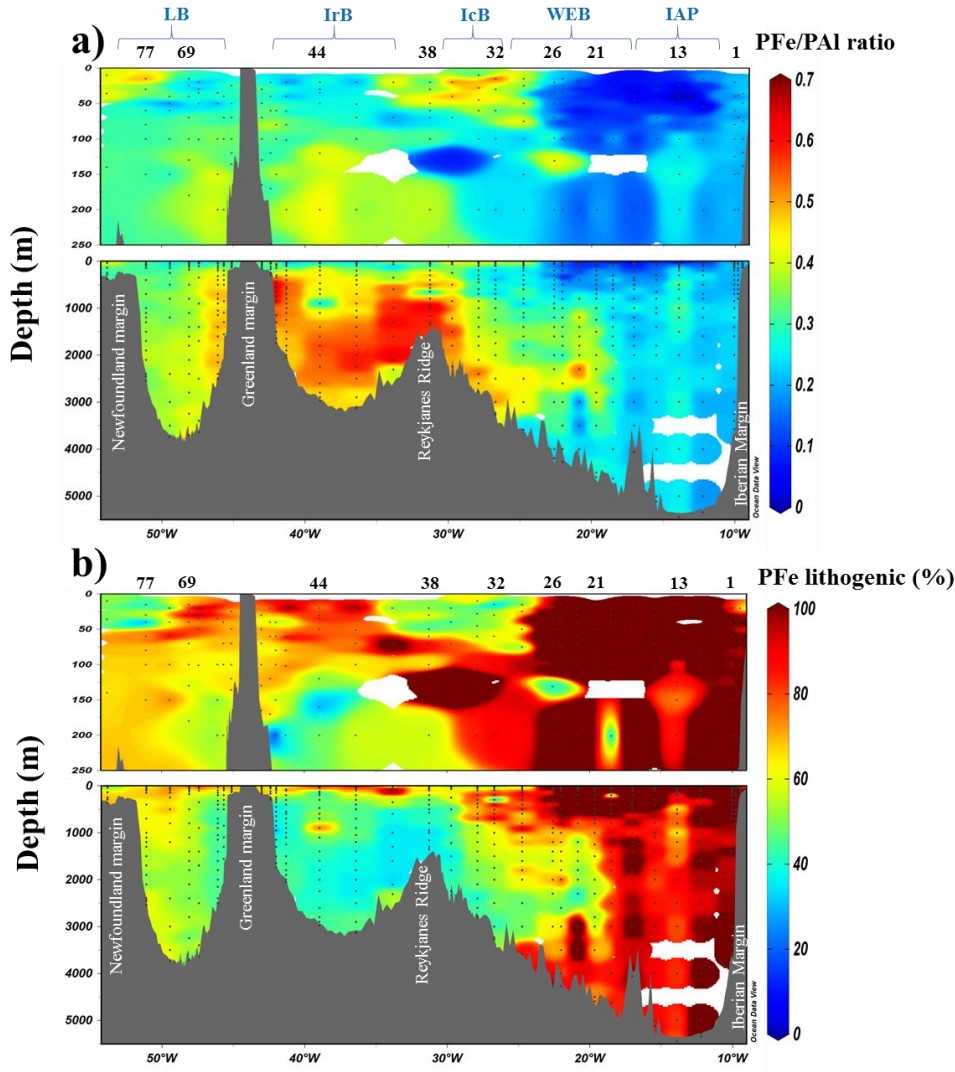










**Figure 7: PFe over PAl concentrations (nmol L⁻¹) for all stations located in the Iberian Abyssal Plain and Western**
**European Basin. Note that the total concentrations of the two elements covaried strongly.**

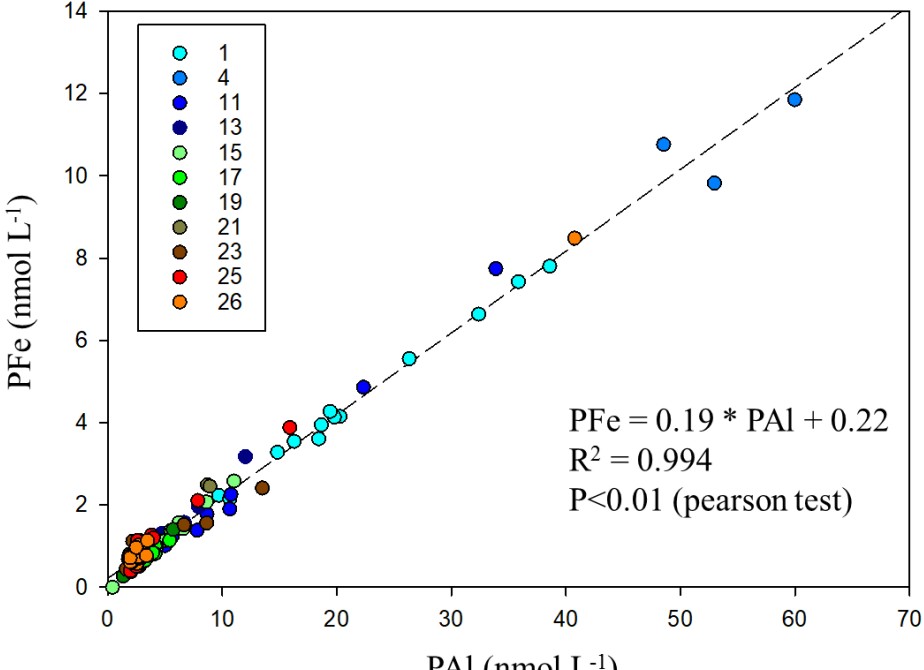


















**Figure 8: Whisker diagram of PFe/PAl molar ratio (mol mol$^{-1}$) in the different water masses sampled along the GA01 line. Median values for the water masses were as follows: LSW$_{lb}$= 0.37; LSW$_{Ir}$=0.44; LSW$_{WEB}$=0.36; ISOW$_{east}$=0.48; ISOW$_{west}$=0.58; DSOW$_{lab}$=0.42; DSOW$_{Ir}$=0.47; NEADW=0.23; MW=0.22 mol mol$^{-1}$.**

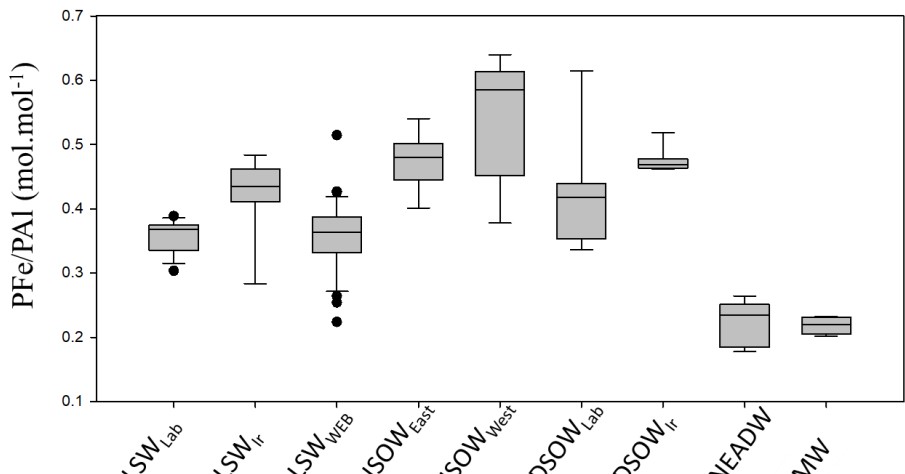



**Figure 9: Scatter of the PFe/PAl ratio at the Iberian (red dots), East Greenland (black dots), West Greenland (green**
**dots) and Newfoundland margins (blue dots). Dashed line indicate the UCC ratio (Taylor and McLennan, 1995).**

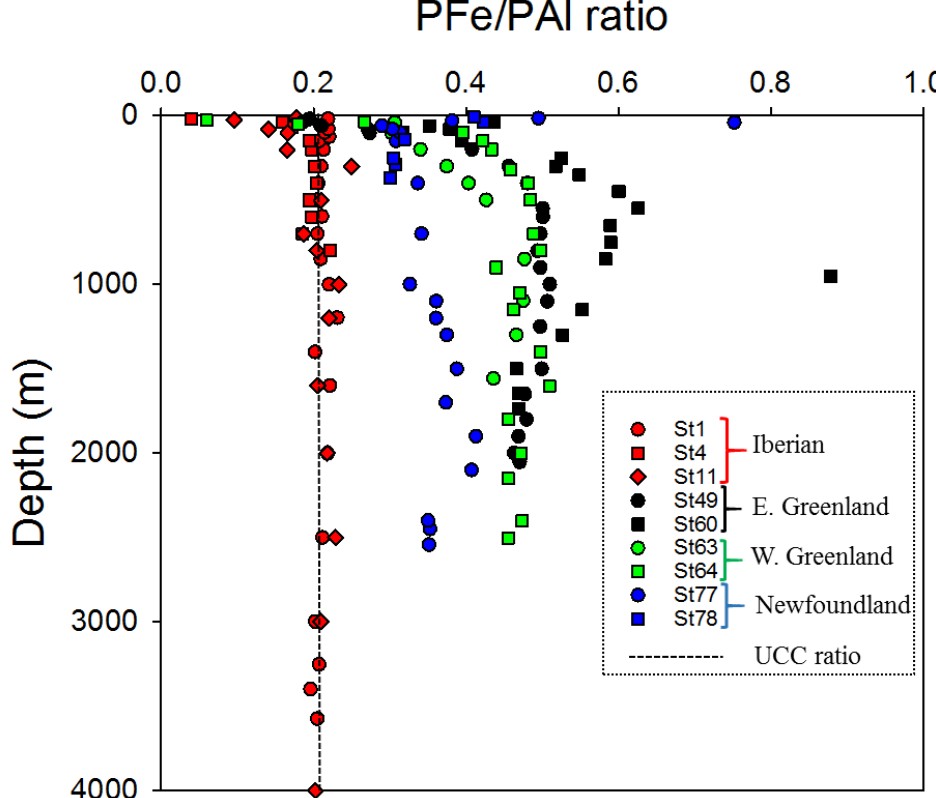













**Figure 10: Section of derived contribution of sedimentary inputs manganese bulk sediment proxy (a) and**
**transmissometry (b) along the GA01 section. Station IDs and biogeochemical region are indicated above the section**
**(a). This figure was generated by Ocean Data View (Schlitzer, R., Ocean Data View, odv.awi.de, 2017).**

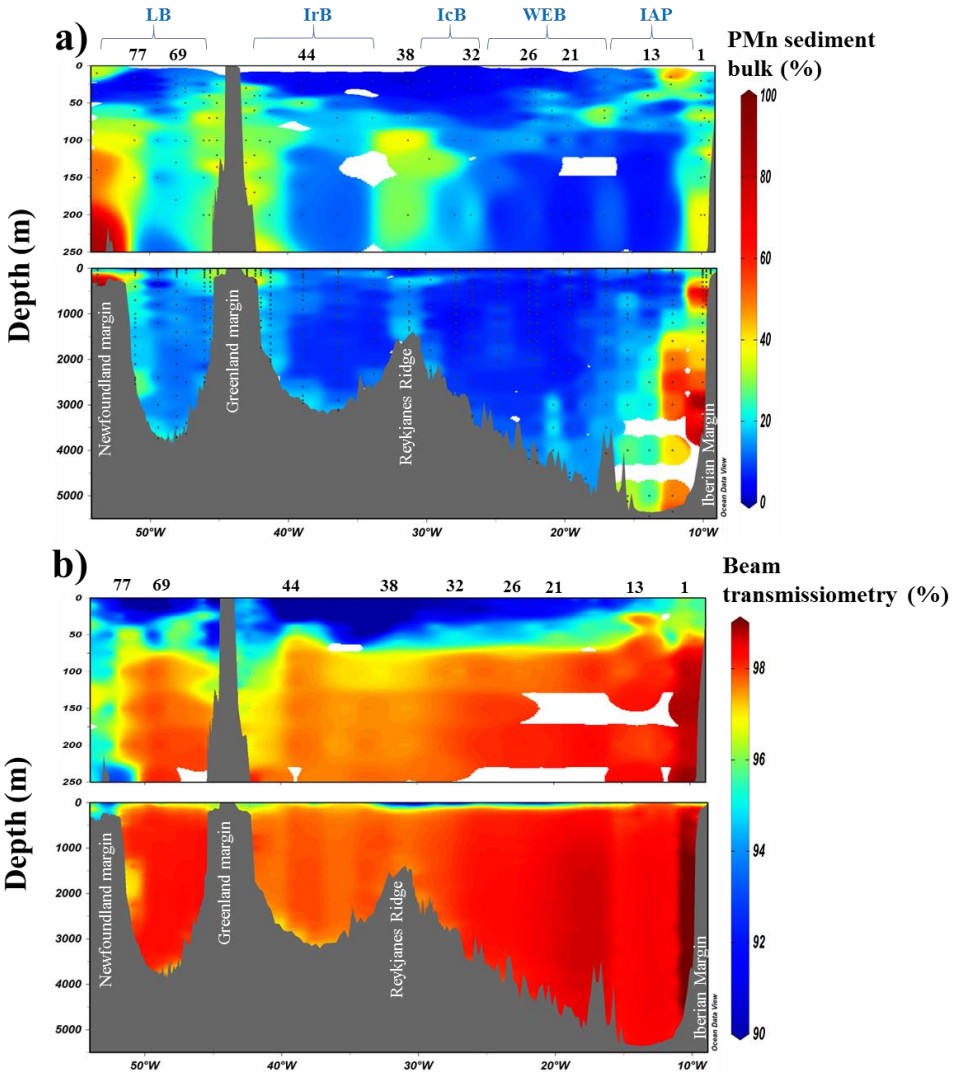









**Figure 11: Vertical profiles of PFe (nmol L⁻¹, a), lithogenic proportion of particulate iron (%, b) and sedimentary**
**proportion of particulate manganese (%, c) at the Iberian, East-West Greenland and Newfoundland margins.**

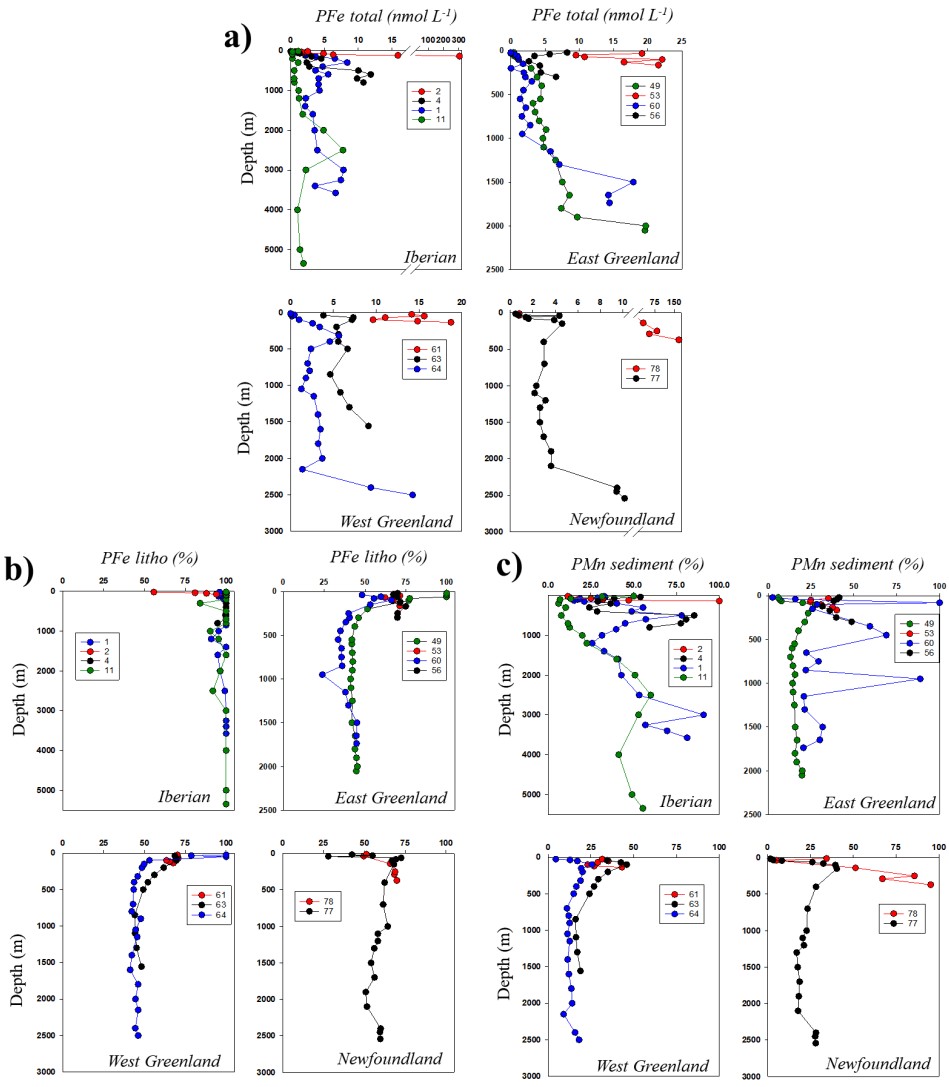












**Figure 12: PFe total (a); PFe/PAl ratio (b) and beam transmissometry (%) as a function of depth above the seafloor**
**(m) at selected stations where a decrease in transmissometry was recorded.**

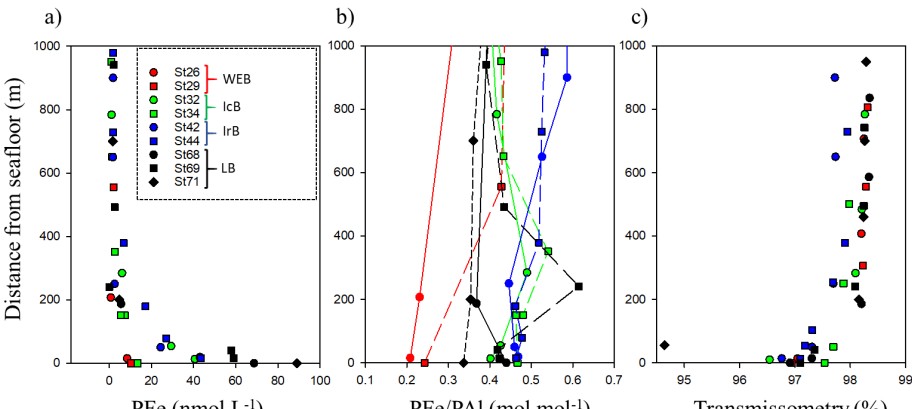






**Figure 13: Vertical profiles of Baxs (grey line, data from Lemaitre et al., 2018a) superimposed with PFe/PAl molar**
**ratios (black dots) at stations sampled in the Iberian Margin (IM), Iberian Abyssal Plain (IAP), Western European**
**Basin (WEB), Iceland Basin (IcB), above the Reyjkanes Ridge (RR), Irminger Basin (IrB), and Labrador Basin (LB).**
**Note that Ba$_{xs}$ concentrations over the background level of 180 pmol L$^{-1}$ are indicative of remineralisation processes**
**(Lemaitre et al., 2018a).**

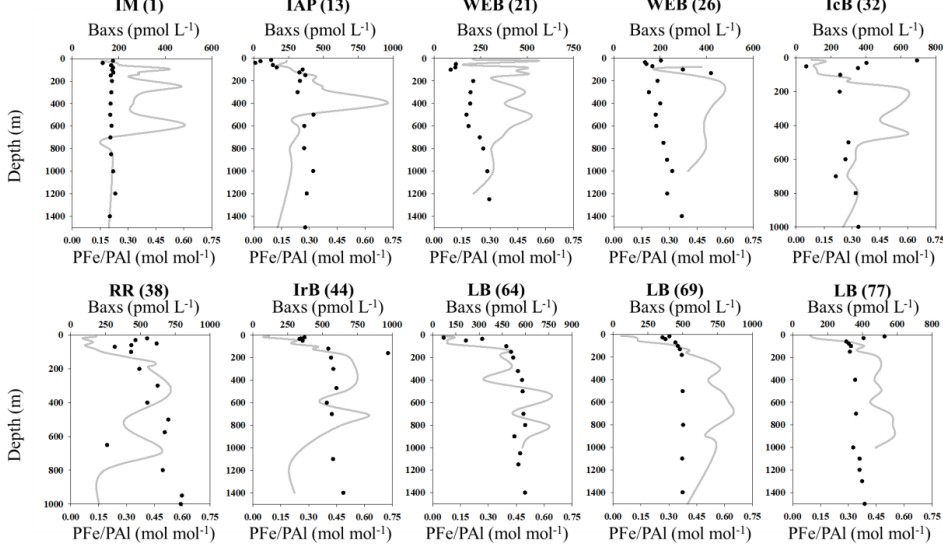







**Figure 14: Vertical profiles of Baxs (grey line, data from Lemaitre et al., 2018a) superimposed with PFe/PP molar**
**ratios (black dots) at stations sampled in the Iberian Margin (IM), Iberian Abyssal Plain (IAP), Western European**
**Basin (WEB), Iceland Basin (IcB), above the Reyjkanes Ridge (RR), Irminger Basin (IrB), and Labrador Basin (LB).**
**Note that Ba$_{xs}$ concentrations over the background level of 180 pmol L$^{-1}$ are indicative of remineralisation processes**
**(Lemaitre et al., 2018a).**

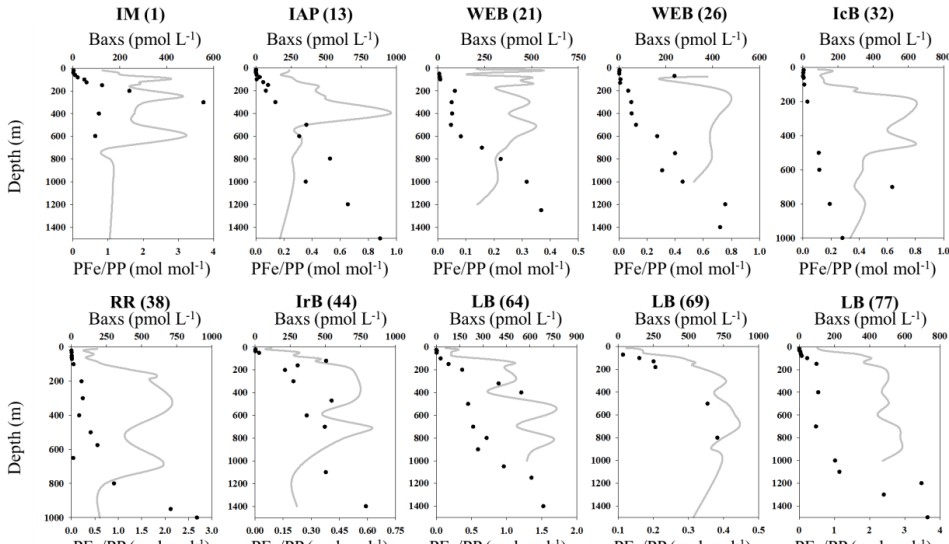











| % recovery | Ba | Al | P | Mn | Fe |
|---|---|---|---|---|---|
| BCR-414 (n=5) | | | | 93 | 94 |
| MESS-4 (n=5) | 110 | 111 | 53 | 104 | 106 |
| PACS-3 (n=4) | | 136 | 104 | 134 | 130 |

**Table 1: Certified reference material (CRM) recoveries during GEOVIDE suspended particle digestion.**






| Author | Year | Fraction | Location | Depth range | PFe | PAl | PMn | PP |
|---|---|---|---|---|---|---|---|---|
| This study | | >0.45μm | N. Atlantic (>40°N) | All | bdl-304 | bdl-1544 | bdl-3.5 | bdl-402 |
| Barrett et al. | 2012 | 0.4um | N. Atlantic (25-60°N) | Upper 1000m | 0.29-1.71 | 0.2-19.7 | | |
| Dammshauser et al. | 2013 | >0.2 μm | Eastern tropical N.A. | 0-200 | | 0.59-17.7 | | |
| Dammshauser et al. | 2013 | >0.2 μm | Meridional Atlantic | 0-200 | | 0.35-16.1 | | |
| Lam et al. | 2012 | 1–51 um | Eastern tropical N.A. | 0-600 | ND-12 | | | |
| Lannuzel et al. | 2011 | >0.2 μm | East Antarctic | Surface | | 0.02-10.67 | 0.01-0.14 | |
| Lannuzel et al. | 2014 | >0.2 μm | East Antarctic | Fast ice | 43-10385 | 121-31372 | 1-307 | |
| Lee et al. | 2017 | >0.8 μm | Eastern tropical S.Pacific | All | bdl-159 | bdl-162 | bdl-8.7 | bdl-983 |
| Marsay et al. | 2017 | >0.4 μm | Ross Sea | All | 0.68-57.3 | ND-185 | ND-1.4 | 5.4-404 |
| Milne et al. | 2017 | >0.45μm | Sub-tropical N.A. | All | ND-140 | ND-800 | | |
| Ohnemus et al. | 2015 | 0.8–51 μm | N. Atlantic | All | 0-938 | 0-3600 | | |
| Planquette et al. | 2009 | >53 μm | Southern Ocean | 30-340 | 0.15–13.2 | 0.11–25.5 | | |
| Schlosser et al. | 2017 | >1 μm | South Georgia Shelf | All | 0.87-267 | 0.6-195 | 0.01-3.85 | |
| Sherrell et al. | 1998 | 1-53um | Northeast Pacific | 0-3557 | | 0.0-54.2 | | |
| Weinstein et al. | 2004 | >53 μm | Labrador Sea | 0-250 | 0.1-1.2 | 0.1-1.5 | | |
| Weinstein et al. | 2004 | 0.4– 10um | Labrador Sea | 0-250 | 2.5 | 3.6 | 0.05 | |
| Weinstein et al. | 2004 | >0.4 μm | Gulf of Maine | 0-300 | 34.8 | 109 | | |


**Table 2: Concentration (in nmol L$^{-1}$) of trace elements (PFe, Pal, PMn and PP) in suspended particles collected in**
**diverse regions of the world's ocean. Bdl: below detection limit, ND: non-determined.**

