# Peer review of "Inputs and processes affecting the distribution of particulate iron in the North Atlantic along the GEOVIDE (GEOTRACES GA01) section"

_Biogeosciences, 2018_

## Referee Comment (RC1) · CS Schlosser (Referee) · 24 Jun 2018

This manuscript presents and discusses the distribution of PFe, PAl, PMn and PP in the high latitudinal North Atlantic. The presented water column data is wonderful and I am looking forward to see the data published in the next GEOTRACES intermediate data product. We need more particulate data! And I really like their PMF calculations. However, the discussion is very detailed and long, but I am missing a straight storyline. The authors jump a lot between different topics and even present Ba data at the end of the manuscript, but a discussion is missing. The manuscript needs serious work, and I am suggesting major revision.

[Figure]

My three main points are:

1. The authors conclude that higher PFe/PAl and PMn/PAl ratios are indicative for biogenic bound particulate Fe. I am missing the discussion of scavenged and authigenic Fe, that could also cause PFe/PAl ratios higher than that of crustal ratios. For my opinion, the authors should include the PFe/PMn ratio, where biogenic ratios (phytoplankton) are available in the literature. 2. The authors include a PMF model and conclude that variances in PFe are related to changes in the content of lithogenic particles. This is in contradiction to the authors conclusion of biogenic Fe, responsible for changes in PFe/PAl. This needs to be discussed more carefully! 3. There is an entire data set of barium excess concentrations at the end of the manuscript. I am not sure that this data is required for the conclusion of the author. If kept, please discuss the data! Detailed comments are listed below. In addition, I have included a pdf conating detailed comments.

With best regards,

Christian Schlosser

Abstract Line 32: What is meant with "At most stations over the Western,..." and "..relative concentration..."? I cannot see how concentrations show a ubiquitous influence of crustal particles. Ratios maybe! However, be more precise.

Introduction Line 78: Replace to "using the distribution of particulate aluminium, manganese, and phosphorous." And remove sub-sentence ", to further..... ."

Methods Line 90: Sentence too long, please split up. Line 91: Missing bracket. Line 97: Indicate Go-Flo company. "General Oceanics" Line 100: 6mm sounds a bit thin for me. Kable must be wider. Line 111: Replace "litters" by "liters" Line 111ff: Filter cleaning and what kind of filters were applied, should be stated earlier. For instance, before how much volume was passed over them. Line 113: Remove "-1" from "M$\Omega$ cm-1" Line 118ff: I cannot follow, is this releant? Line 120: Replace "slide" by "dish"

Line 143: Please provide the values of blanks and limit of detection, maybe in Table 1 (Please also provide the standard deviation of your crm analysis).

Results General comment 1: The result section need to be shortened. You mention in line 277 that PAl and PMn and PFe are similar in IrB, IcB , WEB and IAP (in line 270ff that the Reykjanes ridge is similar to IcB). That is the entire stretch between Spain and Greenland! Please combine results! In addition, if I look at Figure 4, the distribution of PFe in LB seams very similar to the concentrations in IcB. I am suggesting to combine the results of open ocean regions and just include separate paragraphs of results from the three margins, Iberian, Greenland and Newfoundland margin. General comment 2: You talk about different surface currents, please include them in Fig. 1 Line 184ff: Could just find ENACW in Figure 2. Please correct text or Figure 2. Line 198: Remove "really" Line 210: IB refers to IrB and IcB? Please mention that. Line 218ff: There are five concentrations for 4 parameters! Line 221: Please include the standard deviation of trace metals and PP hosted by small particles Line 226: Please refer to transmissometry Figure. Line 228ff: Sentence is hard to follow, please rephrase. Line 233ff: Sentence "The highest..." does not tell anything new, remove! Since you explain results from the Iberian Margin, later referred as (IM), please include IM in Figure 2. Line 240ff: There is something wrong with that sentence! Line 242: When it is really the case at "every stations" then there are no exceptions! Please rephrase. Line 244: I do not understand what is meant here: "Particulate aluminium profiles matched the PFe profiles, with low median concentrations within the first 100m of 1.77 nmol L-1 and 26 pmol L-1 respectively. Then, concentrations increased with depth to reach a maximum close to the oceanic floor." Did you mean 1.77nM Pal and 26pM PFe? Please provide values for bottom waters. Line 258: Replace "progressive" by "gradual". Again refer to transmissometry figure. Line 315: You are mentioning lithogenic elements here. How do you now? I know concentrations are high, but before introducing your tool that differentiate between biogenic and lithogenic Fe, I would leave out such terms.

Discussion Line 323: I would also include run-off, which is probably similar to your

"melting ice shelfs" but more precise. In addition, why sea ice must have melted recently to be a source for PFe. And what do you mean with biological pool? However, please be careful what you state here as source, for instance, lateral mixing is not per se a source, just when PFe loaded waters are advected offshore. Please be more precise! Line 350ff: How barite formation refers to remineralistaion of PFe. Expalin! In addition, what inputs and processes are discussed below! Line 365: Equation 2: I am pretty sure that the * should be -. Line 367: Another possibility might be that lithogenic particles from the Iberian shelf are advected offshore. In addition, the NAC is located further west [D J Reynolds et al., 2016]. Authors need to come up with a better idea, than dust! Line 375ff: This paragraph needs an overhaul! From fronts, via isobaths and isotherms (not shown) to blooms and LSW. It is really hard to follow this paragraph. In general I would have wished the authors explained differences in PFe/PAl ratio and PFe lith% over the entire transect and not just WEB and IAP. Line 384ff: Figure 8: The approach fingerprinting water masses with trace metals such as Fe and Mn would be nice, if it actually works. Other than NEADW and MW, other water masses have a higher Fe/Al ratio but they are very variable. In this case it is vital to check that the water mass difference is significant. I am suggesting performing a student t-test! Line 400-409: You just repeat yourself, please remove! Line 414ff: I am not convinced that the different Fe/Al ratio is driven by different sediments. Where are the elemental ratios of the sediments, just because a sediment is muddy does not proof anything. Further Shelley et al. (2017) showed that dust particles along the GA01 section are mainly from the higher latitudes and not from North Africa. Later on you mention biogenic contribution, you have not introduced this term, and now everything higher than the 0.21 is biogenic. This is questionable, what is with scavenging, authigenic FeOOH formation ect. Line 425: I am not convinced that using a Mn/Al ratio from the upper crust, is helpful tracking sediments. Sediments can have a much higher ratio then the upper crust (eg. Sediments on the shelf of South Georgia 0.0066, Schlosser et al 2018 ). A higher Mn/Al ratio would change your figure 10 entirely. I am suggesting to apply the Mn/Al ratio of sediments from the different regions. In addition, the transmissometry

data in figure 10 need to be cited earlier! Line 490: I am again not convinced that just biological uptake was responsible for elevated Fe/Al ratios. Scavenging and authigenic precipitation would do the same job. You need SEM data to convince me! Line 547ff: What do you mean with "oxido-reductive transformation". I know this term from microbiology classes, but in sediments? I am agreeing, dead biology sinks and settles on the seafloor. However, organic material is quickly remineralised and released Fe will oxidize quickly forming oxyhydroxides. FeOOH precipitate as single particle or form a coating around sediment particles. This will increase your Fe/Al ratio too, and I think even more pronounced than biogenic Fe, which in comparison to lithogenic particles stores just a small amount of Fe. This small quantity will be strongly obscured by lithogenic Fe. Everything below line 548 is highly speculative. If you would like to track biogenic Fe, you should use the Fe/Mn ratio, ratios are provided by T-Y Ho et al. [2003]. His Fe/Mn ratio for phytoplankton is $\sim$ 1.7, lithogenic particles have a significantly higher ratio (upper crust $\sim$50 and sediments $\sim$ 70), indicative for the formation of authigenic Fe. Line 620ff: Any explanation for Fe depleted particles and aerosols, respectively? Anthropogenic? Line 636: I would check the Fe/Mn ratio too! Line 650: Now we are back to PFe/PP and not anymore PFenonlith/PP. This is all very confusing. It is an interesting approach using PP, but what are numbers actually tell us. It would be better to show first how much nonlithogenic Fe is in the top 100 m and plot PP as well in a diagram. By looking at picture 3. PP is similarly high in IrB and LB, changes in PFe/PP are then mainly driven by PFe, but what does it actually mean. Further on, you show nice plots using Ba exess data (Fig. 13 & 14), but there is not a single word towards the end of the discussion. There is more work needed!!

References Ho, T.-Y., A. Quigg, Z. V. Finkel, A. J. Milligan, K. Wyman, P. G. Falkowski, and F. M. M. Morel (2003), The elemental composition of some marine phytoplankton, J. Phycol., 39, 1145-1159. Reynolds, D. J., et al. (2016), Annually resolved North Atlantic marine climate over the last millennium, Nature Communications, 7, 13502.

Please also note the supplement to this comment:

https://www.biogeosciences-discuss.net/bg-2018-234/bg-2018-234-RC1-supplement.pdf

---

## Referee Comment (RC2) · Anonymous Referee #2 · 4 Jul 2018

This manuscript present the vertical distribution of particulate Fe, Al, Mn and P in the North Atlantic along the Geovide section. Particulate trace elements data are still very scarce, and this dataset constitutes a major contribution to our understanding of the biogeochemical cycles of these elements. I am aware that an important work has been done to acquire such a dataset (more than 500 samples!). However, this manuscript is too detailed and the reader can be easily lost. It is difficult to retain clear conclusions from each section. Overall, I think that the discussion section is too ambitious, and the sections about the sources (e.g. dust inputs) and processes (e.g. remineralization) affecting the PFe distribution are sometimes too speculative. The discussion could be improved by adding additional information/parameters collecting during the cruise

(Chl-a, DFe, . . .), and a link between the particulate and dissolved concentrations is missing. The main part of this study used the PFe/PAl ratio to quantify the lithogenic PFe fraction and deduce the non-lithogenic fraction. However, it is likely that this crustal signature is not constant over the Geovide transect. The relevance and limitations of using an unique ratio need to be discussed. This work deserves to be published in Biogeosciences, but only after major revisions (see my comments below).

Specific comments Overall, the introduction and methods are well written. Figures and tables are not enough used in the text to discuss the results. The results section should be shortened – describing the particulate concentrations station by station in is probably not the easiest way to present this dataset. I think the sections 3.2 to 3.10 should be merged and synthetized. In addition, the authors try to describe and explain each feature of the transect. It is probably too ambitious and not so useful. Finally, the size fractionation represents an important information. This aspect is not enough discussed in the manuscript.

L33 – near-ubiquitous . . . but only in the western part of the transect. The sentence is confusing. L36 – I would prefer to see a flux here instead of a concentration. L61 – The term remineralization usually refers to PFe, not DFe. L209-216 –I would remove this section (ms too long), and add one or two sentences with references in the discussion if needed. If this section is conserved, type 6 and 8-haptophytes should be explained. Section 3.3 – A figure or table should be cited to help the reader. Section 3.4 – Once again, Fig. 3 should be cited to help the reader. L330-340 – I would transfer this paragraph in the Methods section. Section 4.1 – This is an interesting approach. I am not sure if it is possible, but it would be very interesting to do such an analysis for two depth horizons, in surface (eg 0-100 m) and below 100 m. It could enable to highlight the vertical distribution of different processes (eg formation of barite mostly in the mesopelagic?). L365 – A term is missing in equation 2. L367-373 – I recommend here to indicate that a biogenic pool is likely present but is masked by the huge proportion of lithogenic PFe. Overall, PFe/PAl is a proxy and the interpretation

should be done with care. L375 – Which feature? The dominance of lithogenic PFe discussed line 369 and 370? L375-383 – This paragraph is a bit confusing. In addition, why only atmospheric inputs are discussed here? L414-416 – This sentence, and the whole paragraph seems to say that the Fe/Al ratio from the UCC used to calculate the lithogenic component is not accurate. I am aware that there is no perfect method to discriminate biogenic and lithogenic Fe and PFe/PAl is only a proxy, however this paragraph clearly contradicts the calculation made before. As it is one of the main objective of the paper, this limitation/bias should be discussed. L416-419 – I may be wrong, but I think that the PFe/PAl signature of the desert dust coming from the Sahara significantly differs from the UCC ratio. See Guieu et al. 2002, Fu et al. 2017, . . . L489 – Replace leaded by led. L502-507 – Other data collected during the cruise could be used here to illustrate the intensity of the bloom. For example, what was the surface chlorophyll a concentration? I recommend to add this kind of information all along the text, it should help making the manuscript less speculative. L533-535 – What does an important phytoplanktonic community mean? It needs to be more precise. Furthermore, a low PFe concentration is not in contradiction with high Chl-a concentrations as usually most of the PFe concentration is from lithogenic or detrital origin and the biogenic pool is usually minor, and driven by intense cycling in surface. L536 – A value / order of magnitude is needed here. Furthermore, it has to be compared with the other areas. L537-539 – This sentence is confusing. L557-564 – To reduce the length of the manuscript, I would remove this paragraph. L552-554 – What did Lam et al. (2017) precisely show? Section 4.3.2 – Here, I cannot see a clear conclusion. L586-601 – This paragraph is probably too long to conclude an absence of hydrothermal inputs. L604 – I can't see these information on Fig. 7. L604-605 – PFe/PAl is higher at station 40 than at station 38. L605 – This a general comment for the whole text: "PMn had a 19% sedimentary origin". The authors refer to a proxy, and should say "about 20%". L616-617 – See my previous comment (L416-419). Section 4.3.4 – Here, there is no clear conclusion. I would recommend to remove this section. L643 – A range of Fe/P cell quotas has been reported for the North Atlantic (see Twining et al.). It would be

interesting here to compare this ratio (assuming 100% of P is from biogenic origin) which gives an estimation of the biogenic PFe in surface with the 100% lithogenic PFe obtained at stations 1-26 using equation 1. This comparison could help to discuss the limitations of such approach. L638-641 – This sentence needs a reference. L646 – Replace pelagic by mesopelagic. L649 – How is defined the remineralization depth? It needs to be explained. L648-650 – PFe/PAl is probably not the best parameter to discuss remineralization since both elements are mostly lithogenic and the variation of this ratio due to remineralization is likely negligible. L650-651 – I am not convinced by this explanation. PP is much more labile than PFe, whatever the remineralization rate. In addition, Fe scavenging could also contributes to this increase in PFe/PP. L652-653 – The authors should explain why scavenging starts to be important only below 600 m depth. L654-659 – This paragraph is confusing. Figure 13 is not introduced and explained. In addition, how the authors conclude to a stronger scavenging of DFe? L661-664 – It is surprising to see a lower remineralization rate for P compared to Fe. This finding should be discussed. In addition, PFe/PP is not presented in a figure and it is hard for the reader to follow the discussion. Section 4.3.5 – Overall, this section is too speculative. The potential impact of the scavenging process is not really discussed, and I think that the use of the PFe/PAl ratio to discuss the different remineralization patterns is not relevant (eg the evolution of DFe would be more appropriate). Finally, it is not easy to draw any clear conclusions form this section. Figures 13 and 14 – These figures are not introduced and discussed in the manuscript. I would remove them and cite the appropriate study instead.

References Fu, Y., Desboeufs, K., Vincent, J., Bon Nguyen, E., Laurent, B., Losno, R., & Dulac, F. (2017). Estimating chemical composition of atmospheric deposition fluxes from mineral insoluble particles deposition collected in the western Mediterranean region. Atmospheric Measurement Techniques, 10(11), 4389-4401. Guieu, C., Loÿe‐Pilot, M. D., Ridame, C., & Thomas, C. (2002). Chemical characterization of the Saharan dust end‐member: Some biogeochemical implications for the western Mediterranean Sea. Journal of Geophysical Research: Atmospheres, 107(D15),

ACH-5.

---

## Author Comment (AC1) · 27 Aug 2018

Dear Dr Schlosser, We would like to thank you for your very constructive comments. All the issues you raised were carefully considered and addressed. Below are our detailed answers, including corresponding lines of text in the revised manuscript. We also took into account reviewer 2 comments, and refer to them accordingly in this response. We provide an updated manuscript with on including all modifications in track changes. We hope that you will find our answers satisfactory and our revised manuscript suitable for publication in this special issue of Biogeosciences Sincerely yours, Arthur Gourain, on behalf of all the authors

"This manuscript presents and discusses the distribution of PFe, PAl, PMn and PP in the high latitudinal North Atlantic. The presented water column data is wonderful and I am looking forward to see the data published in the next GEOTRACES intermediate data product. We need more particulate data! And I really like their PMF calculations. However, the discussion is very detailed and long, but I am missing a straight storyline. The authors jump a lot between different topics and even present Ba data at the end of the manuscript, but a discussion is missing. The manuscript needs serious work, and I am suggesting major revision." We thank the reviewer for his constructive remarks and his acknowledgment of performed work. Regarding the Ba data, we are not presenting them anymore, as we thought the discussion was easier to follow without remineralisation section. "My three main points are: The authors conclude that higher PFe/PAl and PMn/PAl ratios are indicative for biogenic bound particulate Fe. I am missing the discussion of scavenged and authigenic Fe, that could also cause PFe/PAl ratios higher than that of crustal ratios. For my opinion, the authors should include the PFe/PMn ratio, where biogenic ratios (phytoplankton) are available in the literature. It is true that a discussion on authigenic Fe was clearly missing, and we would like to thank you for this very constructive comment. We, unfortunately, cannot investigate specifically this important fraction of particulate iron. We are considering the PFe/PMn is having more bias than PFe/PAl due to the high kinetic of oxidation of Mn within the ocean. Kinetic wich is different than the kinetic of Fe. We now discuss more carefully on the impact of authigenic particles having on PFe/PAl. The authors include a PMF model and conclude that variances in PFe are related to changes in the content of lithogenic particles. This is in contradiction to the authors conclusion of biogenic Fe, responsible for changes in PFe/PAl. This needs to be discussed more carefully! The main lithogenic variability of the PFe is not in contradiction with the biogenic contribution to the PFe/PAl ratio. Indeed with the PMF, we describe the overall variability of PFe. The main variability of the PFe happened with inputs of PFe. These inputs are as described mainly lithogenic and imply the results observed by the PMF. Variation of PFe/PAl ratio between the basins is mainly occurring in open ocean samples where

[Figure]

PFe concentrations are around 1nM. Thus, small variation of PFe at a sub-nanomolar level will highly impact the PFe/PAl elemental ratio. We add some details at the line 293 to avoid any ambiguity of the PMF interpretation. "The PMF analysis has been realised on the entire dataset, in consequence, the factors described are highly influenced by the major variations of particulate element concentrations (at the interface, i.e. margin, seafloor, surface,...)." There is an entire data set of barium excess concentrations at the end of the manuscript. I am not sure that this data is required for the conclusion of the author. If kept, please discuss the data! We removed this section following your advice. Abstract Line 32: What is meant with "At most stations over the Western" and "..relative concentration.."? I cannot see how concentrations show a ubiquitous influence of crustal particles. Ratios maybe! However, be more precise. We modified the sentence as follow: Within the Iberian Abyssal Plain, ratio of PFe over particulate aluminium (PAl) is identical to the continental crust ratio (0.21), indicating the important influence of crustal particles in the water column. Introduction Line 78: Replace to "using the distribution of particulate aluminium, manganese, and phosphorous." And remove sub-sentence ", to further..." Done. Methods Line 90: Sentence too long, please split up. Done. Line 91: Missing bracket. Corrected. Line 97: Indicate Go-Flo company. "General Oceanics". Done. Line 100: 6mm sounds a bit thin for me. Kable must be wider. Our mistake, it was 14mm. Line 111: Replace "litters" by "liters". Done. Line 111ff: Filter cleaning and what kind of filters were applied, should be stated earlier. For instance, before how much volume was passed over them. This sentence has been moved at the beginning of the paragraph (line 113). Line 113: Remove "-1" from "M$\Omega$ cm-1" Done. Line 118ff: I cannot follow, is this releant? It is relevant, as remaining seasalts can seriously reduce the sensitivity of SF-ICP-MS analyses, so they must be reduced as much as possible. Line 120: Replace "slide" by "dish". We did not replace this, as the Millipore company sells these items as Âń Petrislides Âż and not Âń Petri dishes Âż. They don't have the same design as standard petri dishes. Line 143: Please provide the values of blanks and limit of detection, maybe in Table 1 (Please also provide the standard deviation of your crm analysis). These have been added to

Table 1. Results General comment 1: The result section need to be shortened. You mention in line 277 that PAl and PMn and PFe are similar in IrB, IcB , WEB and IAP (in line 270ff that the Reykjanes ridge is similar to IcB). That is the entire stretch between Spain and Greenland! Please combine results! In addition, if I look at Figure 4, the distribution of PFe in LB seams very similar to the concentrations in IcB. I am suggesting to combine the results of open ocean regions and just include separate paragraphs of results from the three margins, Iberian, Greenland and Newfoundland margin. The results have been re-arranged following your advice, with a first section regrouping all open ocean stations, then a section on margins. General comment 2: You talk about different surface currents, please include them in Fig. 1. During the preparation of the manuscript, we tried different possibility to produce a map of our section. Including some currents has been explored. But having the position of every station, their IDs and the different biogeochemical provinces, plus the current make the Figure 1 extremely unclear and difficult to understand. We reference at the beginning of the Hydrography section the paper of Garcia-Ibanez (Line 182). In this paper, the current are well explained and a really good map of the current, their figure 1, is produced.

Line 184ff: Could just find ENACW in Figure 2. Please correct text or Figure 2. The text has been corrected; the water mass is called NACW and not ENACW. Line 198: Remove "really" Done. Line 210: IB refers to IrB and IcB? Please mention that. The IB refers to the Iceland Basin.This paragraph has been deleted in light of the Reviewer comments. Line 218ff: There are five concentrations for 4 parameters! The additional concentration referring to another element finally non-discussed has been removed. It was "21.5"nM. Line 221: Please include the standard deviation of trace metals and PP hosted by small particles Done. Line 226: Please refer to transmissometry Figure. Done. Line 228ff: Sentence is hard to follow, please rephrase. The sentence has been change to: "Within the first 50m, PFe concentrations decreased towards the shelf break where PFe dropped down from 2.53 nmol L-1 (station 2) to 0.8 nmol L-1 (Station 1). Line 248. Line 233ff: Sentence "The highest. . ." does not tell anything new, remove! Since you explain results from the Iberian Margin, later referred as (IM), please include IM in Figure 2. The sentence has been removed. And the figure 2 has been updated, the two other shelves has been also added to keep some consistence. Line 240ff: There is something wrong with that sentence! The sentence has been completely rewritten in light of your General Comment 1. Line 242: When it is really the case at "every stations" then there are no exceptions! Please rephrase. Done. Line 244: I do not understand what is meant here: "Particulate aluminium profiles matched the PFe profiles, with low median concentrations within the first 100m of 1.77 nmol L-1 and 26 pmol L-1 respectively. Then, concentrations increased with depth to reach a maximum close to the oceanic floor." Did you mean 1.77nM Pal and 26pM PFe? Please provide values for bottom waters. The sentence has been completely rewritten in light of your General Comment 1. Line 258: Replace "progressive" by "gradual". Again refer to transmissometry figure. The sentence has been completely rewritten in light of your General Comment 1. Line 315: You are mentioning lithogenic elements here. How do you now? I know concentrations are high, but before introducing your tool that differentiate between biogenic and lithogenic Fe, I would leave out such terms. The word "lithogenic" elements have been removed. Discussion Line 323: I would also include run-off, which is probably similar to your "melting ice shelfs" but more precise. In addition, why sea ice must have melted recently to be a source for PFe. And what do you mean with biological pool? However, please be careful what you state here as source, for instance, lateral mixing is not per se a source, just when PFe loaded waters are advected offshore. Please be more precise! The sentence has been modified to answer the comment as follows: "Possible candidate sources of PFe include lateral advection offshore from the different margins, atmospheric inputs, continental run-off, melting ice shelves and icebergs, resuspended sediments, hydrothermal inputs and biological uptake.". Line 271. Line 350ff: How barite formation refers to remineralistaion of PFe. Expalin! In addition, what inputs and processes are discussed below! A sentence has been added in the previous paragraph at the line 286. The sentence on in the inputs and processes has been removed for more clarity. Line 365: Equation 2: I am pretty sure that the * should be -. Indeed this was a mistake. It has been corrected.

Line 367: Another possibility might be that lithogenic particles from the Iberian shelf are advected offshore. In addition, the NAC is located further west [D J Reynolds et al., 2016]. Authors need to come up with a better idea, than dust! The sentence has been re-written. The combination of the Iberian Margin input and local circulation is now discussed. Line 317. Line 375ff: This paragraph needs an overhaul! From fronts, via isobaths and isotherms (not shown) to blooms and LSW. It is really hard to follow this paragraph. In general I would have wished the authors explained differences in PFe/PAl ratio and PFe lith% over the entire transect and not just WEB and IAP. We completely rewrote this paragraph with more explanation on the front and the change of %PFelitho proxy over the section. Line 321. Line 384ff: Figure 8: The approach fingerprinting water masses with trace metals such as Fe and Mn would be nice, if it actually works. Other than NEADW and MW, other water masses have a higher Fe/Al ratio but they are very variable. In this case it is vital to check that the water mass difference is significant. I am suggesting performing a student t-test! The fingerprints analysis does not work using the Fe and Mn concentrations. We added the statistical analysis done on this. We operate a Kruskal Wallis Anova on-ranks test. This analysis demonstrates the significant difference between the different clusters of stations. We added more detail in the text, lines 330. Line 400-409: You just repeat yourself, please remove! We deleted most of this paragraph except the following sentences: "Inputs from continental shelves and margins have been demonstrated to support high productivity in shallow coastal areas. Inputs of iron from continental margin sediments supporting the high productivity found in shallow coastal regions have been demonstrated in the past (e.g. Cullen et al. (2009), Elrod et al. (2004), Jeandel et al. (2011), Ussher et al. (2007)) and sometimes, were shown to be advected at great distances from the coast (e.g. Lam et al., 2008).". The importance of the margin inputs for biology has not been discussed previously in the paper and we consider it important to mention it. Line 414ff: I am not convinced that the different Fe/Al ratio is driven by different sediments. Where are the elemental ratios of the sediments, just because a sediment is muddy does not proof anything. Further Shelley et al. (2017) showed that dust particles along the GA01 section are mainly from the higher latitudes and not from North Africa. Later on you mention biogenic contribution, you have not introduced this term, and now everything higher than the 0.21 is biogenic. This is questionable, what is with scavenging, authigenic FeOOH formation etc. Throughout this paragraph, we are comparing sediment resuspension at the three different margins. The different mineralogical compositions of these sediments highly influence the composition of the resuspended particles measured. The sediment composition is thus, in our opinion, the key factor influencing the elemental ratio of particles sampled under these conditions (sedimentary resuspension). Unfortunately, we do not have sediment cores at these stations to assess by SEM or chemically their exact composition. Concerning the influence of Saharan dust over the Iberian Margin, Shelley et al. (2015) confirm their influence in the WEB. The influence of northern dust is highlighted in the northern part of the section. You are right that we now refer everything higher than 0.21 as "non-lithogenic" and not "biogenic". Line 425: I am not convinced that using a Mn/Al ratio from the upper crust, is helpful tracking sediments. Sediments can have a much higher ratio then the upper crust (eg. Sediments on the shelf of South Georgia 0.0066, Schlosser et al 2018 ). A higher Mn/Al ratio would change your figure 10 entirely. I am suggesting to apply the Mn/Al ratio of sediments from the different regions. In addition, the transmissometry data in figure 10 need to be cited earlier! We completely agree with your comment, the ideal solution will be to have the exact composition of the sediments at our respective stations. Unfortunately as explained in the previous comment. We don't have access to this information. The different study at our margin (Iberian (Blasco et al., 2000; Merinero et al., 2008) or Greenland (Loring et al., 1996) or in the North Atlantic (Menendez et al., 2017) show important variations of ratio on small spatial range. In consequence, we decided to use a uniform reference value, the UCC value, to compare our different samples. We added a sentence to discuss about the caveats of this proxy. "This proxy assumes homogeneity of the sediment PMn/PAl ratio through the section which is maybe not completely the case at every station. In consequence, this proxy is only a tool to identify new benthic resuspension at specific location and inter-comparison between several locations is not possible." Line370. Line 490: I am again not convinced that just biological uptake was responsible for elevated Fe/Al ratios. Scavenging and authigenic precipitation would do the same job. You need SEM data to convince me! The SEM picture, figure 1, was taken at Station 53, 165m depth.On this image, you can see diatom debris and lithogenic particles and no Fe oxides could be detected on these samples. Line 547ff: What do you mean with "oxido-reductive transformation". I know this term from microbiology classes, but in sediments? I am agreeing, dead biology sinks and settles on the seafloor. However, organic material is quickly remineralised and released Fe will oxidize quickly forming oxyhydroxides. FeOOH precipitate as single particle or form a coating around sediment particles. This will increase your Fe/Al ratio too, and I think even more pronounced than biogenic Fe, which in comparison to lithogenic particles stores just a small amount of Fe. This small quantity will be strongly obscured by lithogenic Fe. Everything below line 548 is highly speculative. If you would like to track biogenic Fe, you should use the Fe/Mn ratio, ratios are provided by T-Y Ho et al. [2003]. His Fe/Mn ratio for phytoplankton is âĹij 1.7, lithogenic particles have a significantly higher ratio (upper crust âĹij50 and sediments âĹij 70), indicative for the formation of authigenic Fe. By "oxido-reductive transformation", we mean any reaction of oxidation and reduction of trace element. These reactions are highly important within the sediment link to the oxygen and sulphur concentration. We agree on the plausible transformation of the biogenic PFe after burial in the sediment. But in the case of a quick oxidation of the dissolved iron from reductive sediments, the speciation of PFe is changing but not the molar ratio between PFe/PAl. Concerning the use of PFe/PMn as a proxy for the biological activity, we think using Mn and not Al brings another uncertainties. Indeed, Mn is highly affected by authigenic formation with different kinetic than iron. We agree that PFe/PAl ratio is not an idealistic proxy but we think it's the best to estimate the contribution of sources over iron cycle. Moreover, results from the PMF demonstrate the weak influence of authigenic particles over the particulate iron cycle. Line 620ff: Any explanation for Fe depleted particles and aerosols, respectively?

Anthropogenic? Buck et al. (2010) are not giving any specific indication about reasons for these particles to be Fe-depleted. Line 636: I would check the Fe/Mn ratio too! This paragraph has been deleted. In light of the reviewers' comments, we decided it was too speculative using the current dataset at our disposition. Line 650: Now we are back to PFe/PP and not anymore PFenonlith/PP. This is all very confusing. It is an interesting approach using PP, but what are numbers actually tell us. It would be better to show first how much nonlithogenic Fe is in the top 100 m and plot PP as well in a diagram. By looking at picture 3. PP is similarly high in IrB and LB, changes in PFe/PP are then mainly driven by PFe, but what does it actually mean. Further on, you show nice plots using Ba exess data (Fig. 13 & 14), but there is not a single word towards the end of the discussion. There is more work needed!! This paragraph has been deleted. In light of the reviewers' comments, we decided it was too speculative using the current dataset

Please also note the supplement to this comment:
https://www.biogeosciences-discuss.net/bg-2018-234/bg-2018-234-AC1-supplement.pdf

[Figure]

| HV | mag □ | mode | WD | spot | dwell | HFW | 20 μm |
|---|---|---|---|---|---|---|---|
| 5.00 kV | 3 000 x | SE | 10.9 mm | 3.0 | 10 μs | 56.9 μm | 412T |

**Fig. 1.**

**Supplement:**

[revised manuscript text omitted]
$^{-1}$ , station ??to 304 nmol L$^{-1}$, Station 2. In sWithin the first 50m, PFe concentrations decreased from the Iberian

Shelf (Station 2, 2.53 nmol L$^{-1}$) towards the shelf break where PFe dropped down from 2.53 nmol  Lnmol L$^{-1}$

(station 2, depth) to 0.8 nmol L$^{-1}$ (Station 1,). PFe concentrations increased with depth at all three stations and reached a maximum at the bottom of station 2 (138.5 m) with more than 300 nmol L$^{-1}$ of PFe.  Lithogenic tracers, such as PAl or PMn, presented similar profiles to PFe with concentrations ranging between 0.11 and

1544 nmol L$^{-1}$, and from below detection limit to 2.51 nmol L$^{-1}$ respectively (station, depth) (Figure 3, Table 1, supplementary material Table S1, Figure xxxxx). Total particulate phosphorus (PP) concentrations were relatively low in surface ranging from undetectable values to 38 nmol L$^{-1}$ (stationxxxxx) (Table 1, supplementary material, Figure xxxx)in surface:, then  Maximum PP was measured in surface at Station 1 (20 m depth), then concentrations decreased with depth and were less than 0.7 nmol L$^{-1}$ below 1000 m depth.

Particulate Fe concentrations inIn the vicinity of the Greenland shelf, PFe concentrations had a high median concentration value of 10.8 nmol L$^{-1}$ (n= ???)whileand were associated with high median PAl and PMn also had high median concentrations of 32.3 nmol L$^{-1}$ (n= ???) and 0.44 nmol L$^{-1}$(n=???), respectively. Concentrations of

PP were high at the surface with a value of 197 nmol L$^{-1}$ at 25 m of station 61. Then, PP concentrations decreased strongly, less than 30 nmol L$^{-1}$, below 100 meters depth. Furthermore, beam transmissometry values in surface waters at these three stations, were the lowest of the entire section, with values below 85 %.

Close to the Newfoundland margin, surface waters displayed a small load of particulate trace metals as PFe,

PAl, and PMn were below 0.8 nmol L$^{-1}$, 2 nmol L$^{-1}$, and 0.15 nmol L$^{-1}$ respectively. Then close to the bottom of station 78, at 371 m, beam transmissometrytry values dropped to 94% and were associated with extremely high concentrations of PFe=168 nmol L$^{-1}$, PAl=559 nmol L$^{-1}$, and PMn=2 nmol L$^{-1}$. Total PP concentrations in the first 50 m ranged from 35 to 97 nmol L$^{-1}$. Below the surface, PP remained relatively high with values up to 16

nmol L$^{-1}$ throughout the water column. (Table 1, supplementary material and Figure XXXX3 and supplementary material Table S1).

*3.5. Reykjanes Ridge (station 38)*

Above the Reykjanes Ridge, the upper portion of the Mid-Atlantic Ridge, particulate trace elements 
[revised manuscript text omitted]
 a less extent atmospheric particles with the North Atlantic Central Water flowing northward (Shelley et al., 2017; Garcia-Ibanez et al., 2015). This point is discussed with more detail in section 4.3.1. While the Iceland, Irminger and the Labrador basins are characterised with median %PFe value under 55%. An interesting feature observable was the dramatic decrease of the %PFe proxy values happening at the station 26 (Figure 6). This feature is likely be associated to the presence of the Sub-Arctic Front, located between 49.5 and 51°N latitude and 23.5 and 22°W longitude (Zunino et al., 2017). Indeed, this front which separates cold and fresh water of subpolar origin from warm and salty water of subtropical origin was clearly identifiable at station 26 by the steep gradient of the isotherms and isohalines (Figure 2).

~~Overall, the lithogenic contribution to PFe varies from 24% (station 60, 950 m) to 100% at stations located within the Western European Basin. This could be linked to a lateral advection of iron rich lithogenic particles sourced from the Iberian margin and to a less extent atmospheric particles with the North Atlantic Central Water flowing northward (Shelley et al., 2017; Garcia Ibanez 
[revised manuscript text omitted]

Overall, the median PFe$_{nonlitho}$/PP molar ratios varied from 1.0 (Irminger Basin) to 38.7 mmol mol$^{-1}$ (Greenland margin) in the upper 50 m. These ratios are consistent with the few available bulk PFe/PP ratios available in the literature (Twining and Baines, 2013 and references therein), ranging from 1 to 31 mmol mol$^{-1}$ and the phytoplankton assemblages encountered during GEOVIDE (Tonnard et al., in prep.). Indeed, the highest PFe$_{nonlitho}$PFe$_{excess}$/PP molar ratio determined at stations 53 and 56 close to the South Greenland margin coincide with a bloom mostly composed of large diatoms, whereas, the smallest ratios were associated with a bloom mainly composed of cyanobacteria and haptophytes (Tonnard et al., in prep.). The effect of biological uptake is also clearly visible when looking at PFe/PAl vertical variation, which increases from the surface to approximately 100m depth (Figure 13), except in the Iberian Margin, which is under the strong influence of lithogenic inputs.

At deeper depths, pelagic remineralisation processes influence the composition of particles (Barbeau et al., 1996, 2001; Boyd et al., 2010; Strzepek et al., 2005). Taking in account remineralization depths that are derived from Baxs proxy which is described and discussed in great detail in Lemaitre et al. (this issue), it is possible to look at the vertical variation of PFebio/PP along the section (Figure 13).

Close to the IM and within the IAP, no PFe/PAl decrease that could point to a preferential remineralisation of PFe over PAl could be observed within the remineralisation depth (200 to 400 m depth, Figure 13), whereas preferential remineralisation of PP over PFe occurs, as reflected by increasing PFe/PP ratios (Figure 14). This is probably due to the fact that remineralisation rates were low (Lemaitre et al., 2018a), and that PFe was mostly of lithogenic origin, more difficult and slow to remineralize (Boyd et al., 2010). Below 600 m depth, scavenging processes could explain the increasing PFe:PP ratios, from 0.30 to 0.80 mol mol$^{-1}$ at station 13, which is consistent with decreasing dFe concentrations within this depth range reported in Tonnard et al. (this issue).

Within the WEB, between 200 and 500 m depth, remineralisation of PFe over PAl occurs, although reported to be small (Lemaitre et al., 2018a) as reflected by decreasing PFe:PAl ratios (Figure 13), while PFe:PP ratios remained constant, pointing out to similar remineralisation rates of PFe and PP. Below 600 m depth, a stronger scavenging of DFe onto particles formation of Fe oxyhydroxydes (si tu peux calculer et reporter les % à ces intervalles de profondeur, ça aiderait) than in IM and IAP is likely to explain the increasing ratios of PFe:PAl from 0.18 to 0.30 mol mol$^{-1}$ and PFe:PP from 0.047 to 0.367 (Station 21), and from 0.16 to 1.05 (Station 26) mol mol$^{-1}$. Similar patterns occur in IcB (station 32), as dFe concentrations increased (Tonnard et al., this issue) therefore ruling out the possibility of PFe enrichment from scavenging.

Above the RR, and in the IrB, at station 38, PFeP is remineralized preferentially over PAl and PPFe, with decreasing increasing 
[revised manuscript text omitted]

---

## Author Comment (AC2) · 27 Aug 2018

Dear reviewer, We would like to thank you for your very constructive comments. All the issues you raised were carefully considered and addressed. Below are our detailed answers, including corresponding lines of text in the revised manuscript. Note that we also took in account Dr Schlosser's comments when we rewrote the manuscript. We also attach the manuscript in track changes as a supplementary material. We hope that you will find our answers satisfactory and our revised manuscript suitable for publication in this special issue of Biogeosciences. Sincerely yours, Arthur Gourain, on behalf of all the authors

[Figure]

This manuscript presents the vertical distribution of particulate Fe, Al, Mn and P in the North Atlantic along the Geovide section. Particulate trace elements data are still very scarce, and this dataset constitutes a major contribution to our understanding of the biogeochemical cycles of these elements. I am aware that an important work has been done to acquire such a dataset (more than 500 samples!). We thank the reviewer for this comment. However, this manuscript is too detailed and the reader can be easily lost. It is difficult to retain clear conclusions from each section. Overall, I think that the discussion section is too ambitious, and the sections about the sources (e.g. dust inputs) and processes (e.g. remineralization) affecting the PFe distribution are sometimes too speculative. We rewrote the discussion in light of this comment and are more cautious with our conclusions. We removed the remineralisation section which was too speculative. The discussion could be improved by adding additional information/parameters collecting during the cruise (Chl-a, DFe, . . .), and a link between the particulate and dissolved concentrations is missing. The link between particulate and dissolved is made and discussed thoroughly in Tonnard et al. (under review for Biogeosciences), together with Chl-a data ; this is why it is not specifically included in the manuscript. More references to Tonnard et al. are included through the discussion. The main part of this study used the PFe/PAl ratio to quantify the lithogenic PFe fraction and deduce the non-lithogenic fraction. However, it is likely that this crustal signature is not constant over the Geovide transect. The relevance and limitations of using an unique ratio need to be discussed. The use of a single PFe/PAl crustal ratio is now discussed line 309. This work deserves to be published in Biogeosciences, but only after major revisions (see my comments below). Specific comments Overall, the introduction and methods are well written. Figures and tables are not enough used in the text to discuss the results. More references to figures and tables are included in the manuscript.

The results section should be shortened – describing the particulate concentrations station by station in is probably not the easiest way to present this dataset. I think the sections 3.2 to 3.10 should be merged and synthetized. This issue was also raised by Dr Schlosser. The results have been re-arranged following your advice, with a first section regrouping all open ocean stations, then a section on margins. In addition, the authors try to describe and explain each feature of the transect. It is probably too ambitious and not so useful. Finally, the size fractionation represents an important information. This aspect is not enough discussed in the manuscript. Regarding the size fractionation, we want to discuss it in a separate paper, which will be focused on the top 100m.

L33 – near-ubiquitous . . . but only in the western part of the transect. The sentence is confusing. The sentence was indeed not clear enough, we changed it by: "Within the Iberian Abyssal Plain, ratio of PFe over particulate aluminium (PAl) is identical to the continental crust ratio (0.21), indicating the important influence of crustal particles in the water column". Line 32. L36 – I would prefer to see a flux here instead of a concentration. A flux will be indeed more interesting but we can't measure a flux over our samples. We're lacking of a spatial resolution to calculate it. L61 – The term remineralization usually refers to PFe, not DFe. Indeed the formulation of the sentence wasn't clear enough, we changed it by: "or produced by remineralisation of particles". Line 61. L209-216 –I would remove this section (ms too long), and add one or two sentences with references in the discussion if needed. If this section is conserved, type 6 and 8-haptophytes should be explained. This section has been removed. Section 3.3 and 3.4 – A figure or table should be cited to help the reader. The Figure 3 is cited at the end of the overview section 3.2 as follow: "Data are shown in Figure 3". This figure includes all the parameters discussed along the following paragraphs. We are now citing this figure throughout this section L330-340 – I would transfer this paragraph in the Methods section. Done. It is now located in the section 2.5, line 166. Section 4.1 – This is an interesting approach. I am not sure if it is possible, but it would be very interesting to do such an analysis for two depth horizons, in surface (eg 0-100 m) and below 100 m. It could enable to highlight the vertical distribution of different processes (eg formation of barite mostly in the mesopelagic?). Indeed this could be interesting to perform and we had a go at it while preparing this manuscript. The main issue we encountered with clustering our dataset by depth range is the loss of positive statistical results. The PMF model needs a lot of data to work properly and by using a small subset of samples, the model is unstable. L365 – A term is missing in equation 2. We modified it. L367-373 – I recommend here to indicate that a biogenic pool is likely present but is masked by the huge proportion of lithogenic PFe. Overall, PFe/PAl is a proxy and the interpretation should be done with care. An additional sentence has been added to explain how the proxies need to be used with care and a comment on biogenic influence has been added from line 309 to 315. L375 – Which feature? The dominance of lithogenic PFe discussed line 369 and 370? The feature described is the dramatic change of regime from station 26. We rearrange this paragraph in light of Dr. Schlosser review. Line 321. L375-383 – This paragraph is a bit confusing. In addition, why only atmospheric inputs are discussed here? We have reworded this paragraph, and added a discussion on the dispersal of Iberian margin rich particles. A similar comment was raised by Dr Schlosser. From line 316 to line 319. L414-416 – This sentence, and the whole paragraph seems to say that the Fe/Al ratio from the UCC used to calculate the lithogenic component is not accurate. I am aware that there is no perfect method to discriminate biogenic and lithogenic Fe and PFe/PAl is only a proxy, however this paragraph clearly contradicts the calculation made before. As it is one of the main objective of the paper, this limitation/bias should be discussed. Regarding this paragraph concerning the benthic inputs of particles, we discussed the different composition of sediments along the section. It is important to not consider sediments as a purely lithogenic source. Benthic sediments are the results of sinking of particles from the above water column. And represent in a certain term, a record of the oceanic particles flux. They are a mix of the overall bulk of particles lithogenic, biogenic and autogenic. Differences of ratio in these sediments are not implying in any way a change of ratio in the crust (continental or oceanic). L416-419 – I may be wrong, but I think that the PFe/PAl signature of the desert dust coming from the Sahara significantly differs from the UCC ratio. See Guieu et al. 2002, Fu et al. 2017, . . . The sentence referring to the aerosol inputs have been removed. L489 – Replace leaded by led. Done. L502-507 – Other data collected during the cruise could be used here to illustrate the intensity of the bloom. For example, what was the surface chlorophyll a concentration? I recommend to add this kind of information all along the text, it should help making the manuscript less speculative. We added the Chl-a concentrations corresponding to the bloom and refer to Tonnard et al. (2018) as the Chl-a data are discussed in this paper. e. L533-535 – What does an important phytoplanktonic community mean? It needs to be more precise. Furthermore, a low PFe concentration is not in contradiction with high Chl-a concentrations as usually most of the PFe concentration is from lithogenic or detrital origin and the biogenic pool is usually minor, and driven by intense cycling in surface. The sentence has been modified in light of this comment "The important phytoplanktonic community present (maximum Chl-a= 4.91 mg m-3, Tonnard et al., in prep), is linked to low PFe of 0.79 nmol L-1 at 10 m, but, with a high PFe/PAl ratio, up to 0.4, and PP concentration of 97 nmol L-1, confirming the biologic influence". Line 472. L536 – A value / order of magnitude is needed here. Furthermore, it has to be compared with the other areas. The sentence has been modified to: "Concerning this latter process, intense remineralization at station 77 (7 mmol C m-2 d-1 compared to 4 mmol C m-2 d-1 in the Western European Basin) has been reported by Lemaitre et al. (2018a),". Line 475. L537-539 – This sentence is confusing. We removed this sentence for clarity purposes. L557-564 – To reduce the length of the manuscript, I would remove this paragraph. We consider that it is important to briefly provide a definition of the benthic nepheloid layers so, to take the reviewer's point on board, we reduced the length of the paragraph as follows: "Benthic nepheloid layers (BNLs) are important layers where local resuspension of sedimentary particles (Bishop and Biscaye, 1982; Eittreim et al., 1976; Rutgers Van Der Loeff et al., 2002) occur due to strong hydrographic stresses (i.e. boundary currents, benthic storms and deep eddies) interacting with the ocean floor ((Biscaye and Eittreim, 1977; Eittreim et al., 1976; Gardner et al., 2017, 2018). Along the GA01 section, BNLs were observable in each province with different strengths (Figures 3 and 12).". Line 494. L552-554 – What did Lam et al. (2017) precisely show? Lam was describing the role of physical characteristic on margin resuspension event. The use wasn't completely appropriate, we removed the sentence. Section 4.3.2 – Here, I cannot see a clear conclusion. We added the following sentence: "Along the GEOVIDE section, BNLs are providing high concentrations of particulate trace element in the deep open ocean that can contribute substantially to the pool of particulate trace elements such as as iron.", Line 516. L586-601 – This paragraph is probably too long to conclude an absence of hydrothermal inputs. The first part of the paragraph has been removed to shorten the paragraph. L604 – I can't see these information on Fig. 7. This paragraph has been removed as explained in the answer of L604-605 comment. L604-605 – PFe/PAl is higher at station 40 than at station 38. Indeed, in light of it, we decided to remove this paragraph due to the lack of significate proof to support this part of the discussion. L605 – This a general comment for the whole text: "PMn had a 19% sedimentary origin". The authors refer to a proxy, and should say "about 20%". This paragraph has been deleted as explained previously. Moreover we've been more careful on the use of proxy over the entire manuscript. L616-617 – See my previous comment (L416-419). We removed the sentence in question. Section 4.3.4 – Here, there is no clear conclusion. I would recommend to remove this section. We want to keep this paragraph about atmospheric inputs. Even if the fact we do not observe any atmospheric deposition is not as interesting as huge deposition events. We think it is important to discuss it, even if the conclusion is not as clear as the other sources. L643 – A range of Fe/P cell quotas has been reported for the North Atlantic (see Twining et al.). It would be interesting here to compare this ratio (assuming 100% of P is from biogenic origin) which gives an estimation of the biogenic PFe in surface with the 100% lithogenic PFe obtained at stations 1-26 using equation 1. This comparison could help to discuss the limitations of such approach. This paragraph has been deleted. In light of the reviewers' comments, we decided it was too speculative using the current dataset at our disposition. L638-641 – This sentence needs a reference. This paragraph has been deleted. In light of the reviewers' comments, we decided it was too speculative using the current dataset at our disposition. L646 – Replace pelagic by mesopelagic. This paragraph has been deleted. In light of the reviewers' comments, we decided it was too speculative using the current dataset at our disposition. L649 – How is defined the remineralization depth? It needs to be explained. This paragraph has been deleted. In light of the reviewers' comments, we decided it was too speculative using the current dataset at our disposition. L648-650 – PFe/PAl is probably not the best parameter to discuss remineralization since both elements are mostly lithogenic and the variation of this ratio due to remineralization is likely negligible. This paragraph has been deleted. In light of the reviewers' comments, we decided it was too speculative using the current dataset at our disposition. L650-651 – I am not convinced by this explanation. PP is much more labile than PFe, whatever the remineralization rate. In addition, Fe scavenging could also contributes to this increase in PFe/PP. This paragraph has been deleted. In light of the reviewers' comments, we decided it was too speculative using the current dataset at our disposition. L652-653 – The authors should explain why scavenging starts to be important only below 600 m depth. This paragraph has been deleted. In light of the reviewers' comments, we decided it was too speculative using the current dataset at our disposition. L654-659 – This paragraph is confusing. Figure 13 is not introduced and explained. In addition, how the authors conclude to a stronger scavenging of DFe? This paragraph has been deleted. In light of the reviewers' comments, we decided it was too speculative using the current dataset at our disposition. L661-664 – It is surprising to see a lower remineralization rate for P compared to Fe. This finding should be discussed. In addition, PFe/PP is not presented in a figure and it is hard for the reader to follow the discussion. This paragraph has been deleted. In light of the reviewers' comments, we decided it was too speculative using the current dataset at our disposition. Section 4.3.5 – Overall, this section is too speculative. The potential impact of the scavenging process is not really discussed, and I think that the use of the PFe/PAl ratio to discuss the different remineralization patterns is not relevant (eg the evolution of DFe would be more appropriate). Finally, it is not easy to draw any clear conclusions form this section. This paragraph has been deleted. In light of the reviewers' comments, we decided it was too speculative using the current dataset at our disposition. Figures 13 and 14 – These figures are not introduced and discussed in the manuscript. I would remove them and cite the appropriate study instead. This paragraph has been deleted. In light of the reviewers' comments, we decided it was too speculative using the current dataset at our disposition.

Please also note the supplement to this comment:
https://www.biogeosciences-discuss.net/bg-2018-234/bg-2018-234-AC2-supplement.pdf

**Supplement:**

[revised manuscript text omitted]
$^{-1}$ , station ??to 304 nmol L$^{-1}$, Station 2. In sWithin the first 50m, PFe concentrations decreased from the Iberian

Shelf (Station 2, 2.53 nmol L$^{-1}$) towards the shelf break where PFe dropped down from 2.53 nmol  Lnmol L$^{-1}$

(station 2, depth) to 0.8 nmol L$^{-1}$ (Station 1,). PFe concentrations increased with depth at all three stations and reached a maximum at the bottom of station 2 (138.5 m) with more than 300 nmol L$^{-1}$ of PFe.  Lithogenic tracers, such as PAl or PMn, presented similar profiles to PFe with concentrations ranging between 0.11 and

1544 nmol L$^{-1}$, and from below detection limit to 2.51 nmol L$^{-1}$ respectively (station, depth) (Figure 3, Table 1, supplementary material Table S1, Figure xxxxx). Total particulate phosphorus (PP) concentrations were relatively low in surface ranging from undetectable values to 38 nmol L$^{-1}$ (stationxxxxx) (Table 1, supplementary material, Figure xxxx)in surface:, then  Maximum PP was measured in surface at Station 1 (20 m depth), then concentrations decreased with depth and were less than 0.7 nmol L$^{-1}$ below 1000 m depth.

Particulate Fe concentrations inIn the vicinity of the Greenland shelf, PFe concentrations had a high median concentration value of 10.8 nmol L$^{-1}$ (n= ???)whileand were associated with high median PAl and PMn also had high median concentrations of 32.3 nmol L$^{-1}$ (n= ???) and 0.44 nmol L$^{-1}$(n=???), respectively. Concentrations of

PP were high at the surface with a value of 197 nmol L$^{-1}$ at 25 m of station 61. Then, PP concentrations decreased strongly, less than 30 nmol L$^{-1}$, below 100 meters depth. Furthermore, beam transmissometry values in surface waters at these three stations, were the lowest of the entire section, with values below 85 %.

Close to the Newfoundland margin, surface waters displayed a small load of particulate trace metals as PFe,

PAl, and PMn were below 0.8 nmol L$^{-1}$, 2 nmol L$^{-1}$, and 0.15 nmol L$^{-1}$ respectively. Then close to the bottom of station 78, at 371 m, beam transmissometrytry values dropped to 94% and were associated with extremely high concentrations of PFe=168 nmol L$^{-1}$, PAl=559 nmol L$^{-1}$, and PMn=2 nmol L$^{-1}$. Total PP concentrations in the first 50 m ranged from 35 to 97 nmol L$^{-1}$. Below the surface, PP remained relatively high with values up to 16

nmol L$^{-1}$ throughout the water column. (Table 1, supplementary material and Figure XXXX3 and supplementary material Table S1).

*3.5. Reykjanes Ridge (station 38)*

Above the Reykjanes Ridge, the upper portion of the Mid-Atlantic Ridge, particulate trace elements 
[revised manuscript text omitted]
 a less extent atmospheric particles with the North Atlantic Central Water flowing northward (Shelley et al., 2017; Garcia-Ibanez et al., 2015). This point is discussed with more detail in section 4.3.1. While the Iceland, Irminger and the Labrador basins are characterised with median %PFe value under 55%. An interesting feature observable was the dramatic decrease of the %PFe proxy values happening at the station 26 (Figure 6). This feature is likely be associated to the presence of the Sub-Arctic Front, located between 49.5 and 51°N latitude and 23.5 and 22°W longitude (Zunino et al., 2017). Indeed, this front which separates cold and fresh water of subpolar origin from warm and salty water of subtropical origin was clearly identifiable at station 26 by the steep gradient of the isotherms and isohalines (Figure 2).

~~Overall, the lithogenic contribution to PFe varies from 24% (station 60, 950 m) to 100% at stations located within the Western European Basin. This could be linked to a lateral advection of iron rich lithogenic particles sourced from the Iberian margin and to a less extent atmospheric particles with the North Atlantic Central Water flowing northward (Shelley et al., 2017; Garcia Ibanez 
[revised manuscript text omitted]

Overall, the median PFe$_{nonlitho}$/PP molar ratios varied from 1.0 (Irminger Basin) to 38.7 mmol mol$^{-1}$ (Greenland margin) in the upper 50 m. These ratios are consistent with the few available bulk PFe/PP ratios available in the literature (Twining and Baines, 2013 and references therein), ranging from 1 to 31 mmol mol$^{-1}$ and the phytoplankton assemblages encountered during GEOVIDE (Tonnard et al., in prep.). Indeed, the highest PFe$_{nonlitho}$PFe$_{excess}$/PP molar ratio determined at stations 53 and 56 close to the South Greenland margin coincide with a bloom mostly composed of large diatoms, whereas, the smallest ratios were associated with a bloom mainly composed of cyanobacteria and haptophytes (Tonnard et al., in prep.). The effect of biological uptake is also clearly visible when looking at PFe/PAl vertical variation, which increases from the surface to approximately 100m depth (Figure 13), except in the Iberian Margin, which is under the strong influence of lithogenic inputs.

At deeper depths, pelagic remineralisation processes influence the composition of particles (Barbeau et al., 1996, 2001; Boyd et al., 2010; Strzepek et al., 2005). Taking in account remineralization depths that are derived from Baxs proxy which is described and discussed in great detail in Lemaitre et al. (this issue), it is possible to look at the vertical variation of PFebio/PP along the section (Figure 13).

Close to the IM and within the IAP, no PFe/PAl decrease that could point to a preferential remineralisation of PFe over PAl could be observed within the remineralisation depth (200 to 400 m depth, Figure 13), whereas preferential remineralisation of PP over PFe occurs, as reflected by increasing PFe/PP ratios (Figure 14). This is probably due to the fact that remineralisation rates were low (Lemaitre et al., 2018a), and that PFe was mostly of lithogenic origin, more difficult and slow to remineralize (Boyd et al., 2010). Below 600 m depth, scavenging processes could explain the increasing PFe:PP ratios, from 0.30 to 0.80 mol mol$^{-1}$ at station 13, which is consistent with decreasing dFe concentrations within this depth range reported in Tonnard et al. (this issue).

Within the WEB, between 200 and 500 m depth, remineralisation of PFe over PAl occurs, although reported to be small (Lemaitre et al., 2018a) as reflected by decreasing PFe:PAl ratios (Figure 13), while PFe:PP ratios remained constant, pointing out to similar remineralisation rates of PFe and PP. Below 600 m depth, a stronger scavenging of DFe onto particles formation of Fe oxyhydroxydes (si tu peux calculer et reporter les % à ces intervalles de profondeur, ça aiderait) than in IM and IAP is likely to explain the increasing ratios of PFe:PAl from 0.18 to 0.30 mol mol$^{-1}$ and PFe:PP from 0.047 to 0.367 (Station 21), and from 0.16 to 1.05 (Station 26) mol mol$^{-1}$. Similar patterns occur in IcB (station 32), as dFe concentrations increased (Tonnard et al., this issue) therefore ruling out the possibility of PFe enrichment from scavenging.

Above the RR, and in the IrB, at station 38, PFeP is remineralized preferentially over PAl and PPFe, with decreasing increasing 
[revised manuscript text omitted]

---

## Referee Report (RR1)

Review of the revised manuscript "Inputs and processes affecting the distribution of particulate Fe in the North Atlantic along the GEOVIDE section" by Gourain, A. and co-authors

First of all the manuscript has improved massively compared to the last version. The first half of the manuscript is mostly in good shape, but the second part lacks mainly a clear structure of paragraphs which makes it really hard to follow the stream of thoughts. So, especially the conclusion needs to be revisited and strengthened. There is a lot of text, but little information. I am also missing the highlighting of the biogenic fraction, there is hardly anything discussed. However, the data is interpreted and concluded correctly, but changes to the text that need to be done to improve the second part of the paper require between major and intermediate revision. I hope my comments below help to strengthen the text.

With best regards,

Christian Schlosser

Abstract

Line 35:replace "basins" by "basin"

Line 36ff: Important sounds strange! I would use "high" instead. Please alos include "horizontal", otherwise advection takes also place vertically. I would also include "advection of PFe containing water masses " not PFe travels the water mass does this job.

Line 40ff: This sentence is a bit lost, and out of context.

Method

Line 89: "briefly described in section 2.1" I cannot find a brief description of the complex circulation.

Line 102; Remove "and the filters processed…" This is out of context here!

Line 104: What find of filters you are talking about, these are not filters used for the Swinnex filtration right?

Line 133: Replace "filter" by "Filters"

Line 137: You also removed the filter, right?

Results

Line 180ff: I have not found any biological settings in this chapter… It is also very long, detailed, and to some extend hard to follow. It would be good to start with a sentence what you are doing and then characterize water masses basin after basin separated by different paragraphs. That would make it easier to follow.

Line 222: This sentence is really hard to understand, please rephrase. Please also shorten the title!

Line 229: I have not even watched Figure 3 and sup. Table 1and you introduce Figure 9b and sup. Table 2.

Line 253: Include "in THE surface"

Line 281: You refer here to 5 factors, figure 6 represents 4 factors. What is right, I presume the figure?

Line 286: change to "and remineralisation of biogenic material."

Line 300: Replace "material" by " fraction".

Line 312: Replace "with consideration" by "carefully".

Line 320ff: with %PFe you mean the lithogenic fraction…, be careful and try to apply always the same abbreviation.

Line 320-325: If keeping these sentence, please create a new paragraph. Please also rewrite sentences, they are really hard to understand and follow!  Link the %PFe data to Salinity, that at least gives you some certainty that there is a change in water mass.

Line 335-340: Please rephrase the sentences, sorry hard to read! Especially the last sentence is formulated  very vague!

Line 344ff: The following 3 paragraphs should have been included already in section 4.2.You are discussing your results and not discussing  them. The structure of the following paragraphs needs to be changed as well, first you come with your hypothesis and then you explain why this is the case. Right now it is all turned around. Why are you coming up with the ratio Mn/Al  now. It comes out of the blue. I would introduce this parameter earlier when you come up with %PFe for instance.

Line 379: This paragraph is not explaining the differences of the Fe/Al ratio observed. What is the River doing?

Line 383:What do you mean with important!

Line 396ff: This sentence needs to come first.

Line 399ff: Your paragraphs are really long. Anyway, you do not discuss why there are this two maxima. The last sentence "Therefore…" is the most important finding and should come first. And then switching between PFe and PMn, it is hard to follow your stream of thoughts.

Line 423ff: What is the message of the paragraph. Buried in the text "Transfer of DFe to PFe. That needs to come in the beginning of the paragraph.

Line 479ff: Good this is the most important paragraph, for the first time you put the finding into context and discuss them. It is very hard to go through the last 3 pages without a strong structured line of thoughts!

Line 494ff: The presented literature data needs to be put in context to your findings towards the end of the following paragraph.

Line 526: Include the SEM picture in the sup material.

Line 535ff: First: Barraqueta, Second: What do you mean with the following sentence. It is hard to follow. What has the concentration to do with the composition?

Conclusion

This is not a conclusion, it is just a second abstract.

Line 550: Maybe start the Sentence with: The

Line 555: The river is not responsible or the PFe at 2500 m depth.

Figure 2 citation: Line 908: Replace "Stations" by "Station"

Figure 3: Please increase the font size of the ODV graphs, it is almost impossible to read the depth and longitude on a print out.

Figure 4 caption: Please include that the PFe scale changes within graphs.

---

## Referee Report (RR2)

Review on "Inputs and processes affecting the distribution of particulate iron in the North Atlantic along the GEOVIDE (GEOTRACES GA01) section" by Gourain et al.

I stopped reviewing the manuscript at line 366. Due to the state of the English language, sentences are difficult, some almost impossible to understand. This needs to be fixed before the next round of review should start. I am suggesting that a native English speaking person has a look and improves the English. Sorry for that, but I need to recommend major revision. However, I am still willing to review the next version, but insist on good English grammar.

Sincerely,

Christian Schlosser

Some minor points:

Line 35: Change to "…(0.21 mol mol-1)…"

Line 38ff: Include the second decimal digit for 0.7 and remove the ratio for the continental crust (you just told it 2 sentences earlier.

Line 42: Maybe "suspending" would be better than "delivering".

Line 51: Here "deliver" would be better than "bring"

Line 65ff: What do you mean with dissolution? Dissolution of inorganic PFe? Is not that also part of the regeneration?

Line 101: Replace "They" by "Bottles"

Line 123: Rewrite sentence "before to pass".

Line 189: Remove "really"

Line 190ff: What do you mean with "These authigenic particles lead to an enrichment of Mn in particle compositions."

Line 195: Replace "of direct and recent" by "for"

Line 226: "modal", do you mean "mode"?

Line 242: Change "Total particulate concentrations spanned a large range of concentrations from below detection to 304 nmol L-1 for PFe, 1544 nmol L-1 for PAl, 3.5 nmol L-1 for PMn and 402 nmol L-1 for PP."

Line 275ff: include "depth" after you introduce the depth of a sample , for example "at 25 m depth" in line 286. Please apply throughout the paragraph/manuscript.

Line 299: Remove "candidate"

Line 300: Replace "ice shelves" by "glaciers"

Line 350: Remove "candidate"

Figure 5: Please include in the legend what the different factors are (eg. lithogenic). Makes it easier to follow!

---

## Referee Report (RR3)

Review on "Inputs and processes affecting the distribution of particulate iron in the North Atlantic along the GEOVIDE (GEOTRACES GA01) section" by Gourain et al.

The manuscript has improved much! I agree with the scientific findings and suggestions made, but still the language is sometimes hard to follow and paragraphs are not always, but sometimes very long. All this makes it to a real challenge to read the manuscript.

From the scientific part, I have just some minor points, listed below

Line 32: Insert "thermohaline overturning circulation."

Line 131: I do not know what the Berger citation has to do with your results. I would remove the Berger et al. citation!

Line 210: Include "Close to the sea floor…."

Line 318: Replace (PMF, factor 3 = 4.1%)

Line 319ff: Replace ", but its contribution is most likely obscured…"

Line 321ff: The reasoning is plausible, but please shorten this paragraph, it is too long. You can distill the message down to a couple sentences.

Line 388: Replace "in" by "to the Atlantic Ocean."

Line 401-409: I do not follow the reasoning! PP is high because of high DFe and elevated primary productivity, which subsequently reduced DFe by biological utilization. First you need to show that DFe concentrations are elevated. Then use Chl a to show that the production is elevated, and then you may argue that the high Pfe with a high PFe/Pal ratio comes from elevated productivity.

Line 410ff: Shorten the paragraph, you say a lot without saying something useful.

Line 446ff: This paragraph is out of context. Also the next one, which I suppose was written to sum up the results. But somehow it doesn't, maybe better to create another section.

Line 466ff: What is new, that has not been said earlier. Nepheloid layers have been already introduced!

Line 490ff: You could also use DFe and DMn as a tracer. What are these elements showing!

Line 527ff: This is speculative!

---

## Author Response (AR3)

Review of the revised manuscript "Inputs and processes affecting the distribution of particulate Fe in the North Atlantic along the GEOVIDE section" by Gourain, A. and co-authors.

First of all the manuscript has improved massively compared to the last version. The first half of the manuscript is mostly in good shape, but the second part lacks mainly a clear structure of paragraphs which makes it really hard to follow the stream of thoughts. So, especially the conclusion needs to be revisited and strengthened. There is a lot of text, but little information. I am also missing the highlighting of the biogenic fraction, there is hardly anything discussed. However, the data is interpreted and concluded correctly, but changes to the text that need to be done to improve the second part of the paper require between major and intermediate revision. I hope my comments below help to strengthen the text.

With best regards,

Christian Schlosser

Dear Dr Schlosser,

We would like to thank you for your time and effort in this review process.

Concerning the biogenic fraction, it is true that while we are mentioning it in our PMF analysis, we do not discuss it much further, as this is the focus of another paper currently under preparation (distribution between size fractions in the top 200m, completed with chemical leaches analyses). We wrote a specific sentence to mention this lines 358-359.

We reorganized most of the discussion, and simplified some of the wordy sentences.

We truly hope that you will now find this manuscript suitable for publication.

Our detailed answers are below.

Kind regards,

Arthur Gourain, on behalf of all coauthors.

**Abstract**

Line 35:replace "basins" by "basin"          Done.

Line 36ff: Important sounds strange! I would use "high" instead. Please alos include "horizontal", otherwise advection takes also place vertically. I would also include "advection of PFe containing water masses " not PFe travels the water mass does this job.          Done.

Line 40ff: This sentence is a bit lost, and out of context.          We removed this sentence.

**Method**

Line 89: "briefly described in section 2.1" I cannot find a brief description of the complex circulation.
          Indeed the circulation is described section 3.1. We changed 2.1 by 3.1 in the text.

Line 102; Remove "and the filters processed…" This is out of context here!          Done.

Line 104: What find of filters you are talking about, these are not filters used for the Swinnex filtration right?
          Yes, we are talking about the Swinnex filtration.

Line 133: Replace "filter" by "Filters"          Done.

Line 137: You also removed the filter, right?          Yes, this is now indicated.

**Results**

Line 180ff: I have not found any biological settings in this chapter… It is also very long, detailed, and to some extend hard to follow. It would be good to start with a sentence what you are doing and thenm characterize water masses basin after basin separated by different paragraphs. That would make it easier to follow.          We removed the biological settings from the title, they were mentioned in the first version of the manuscript and we omit to remove them from the title. We also created more paragraphs to make it easier to follow.

Line 222: This sentence is really hard to understand, please rephrase. Please also shorten the title!          We shortened the title as requested to "Open Ocean stations: from the Iberian Abyssal Plain to the Labrador Basin and add an introductory line to precise which stations are concerned by this paragraph. Line 254.

We modified the sentence as follow: "Particulate iron concentration profiles showed identical patterns at all of the open ocean stations encountered along the section."

Line 229: I have not even watched Figure 3 and sup. Table 1and you introduce Figure 9b and sup. Table 2.          We removed the citation.

Line 253: Include "in THE surface"          Done.

Line 281: You refer here to 5 factors, figure 6 represents 4 factors. What is right, I presume the figure?          Indeed 4 factors is the right one.

Line 286: change to "and remineralisation of biogenic material."          Done.

Line 300: Replace "material" by " fraction".          Done.

Line 312: Replace "with consideration" by "carefully".          Done.

Line 320ff: with %PFe you mean the lithogenic fraction…, be careful and try to apply always the same abbreviation.          Indeed there was a typo mistake. It's been modified to %PFe$_{litho.}$

Line 320-325: If keeping these sentence, please create a new paragraph. Please also rewrite sentences, they are really hard to understand and follow! Link the %PFe data to Salinity, that at least gives you some certainty that there is a change in water mass.          We rewrote this paragraph and included salinity values to demonstrate the presence of the front and thus confirm the link between %PFelitho and salinity value. Lines 344 to 346.

Line 335-340: Please rephrase the sentences, sorry hard to read! Especially the last sentence is formulated very vague!          Done

Line 344ff: The following 3 paragraphs should have been included already in section 4.2.You are discussing your results and not discussing them. The structure of the following paragraphs needs to be changed as well, first you come with your hypothesis and then you explain why this is the case. Right now it is all turned around. Why are you coming up with the ratio Mn/Al now. It comes out of the blue. I would introduce this parameter earlier when you come up with %PFe for instance.          We are now introducing %PFe litho and %PMn in the methods (section 2.6) as derived parameters. This way, the flow of the discussion is not altered by the explanation of these parameters and flows better. Furthermore, we deleted the subsection title "fingerprinting water masses" and merged the discussion of the water masses exhibiting a specific PFe/PAl with the PMF results and the %PFe litho (lines 380-381; ).

Line 379: This paragraph is not explaining the differences of the Fe/Al ratio observed. What is the River doing?

We deleted this paragraph. We were not discussing the PFe/PAl ratio from the water discharge due to the lack of data within the plume and the endmember. The differences of PFe/PAl ratio described line 472 are linked to the margin sediments resuspension.

Line 383:What do you mean with important!          We replaced important by high.

Line 396ff: This sentence needs to come first.          We reorganized this paragraph.

Line 399ff: Your paragraphs are really long. Anyway, you do not discuss why there are this two maxima. The last sentence "Therefore…" is the most important finding and should come first. And then switching between PFe and PMn, it is hard to follow your stream of thoughts.          We reorganized this paragraph.

Line 423ff: What is the message of the paragraph. Buried in the text "Transfer of DFe to PFe. That needs to come in the beginning of the paragraph.          We reorganized this paragraph.

Line 479ff: Good this is the most important paragraph, for the first time you put the finding into context and discuss them. It is very hard to go through the last 3 pages without a strong structured line of thoughts!          Thanks, as previously explained, we rewrote the previous paragraph in light of your comments.

Line 494ff: The presented literature data needs to be put in context to your findings towards the end of the following paragraph.          We reorganized this paragraph.

Line 526: Include the SEM picture in the sup material.          Done.

Line 535ff: First: Barraqueta, Second: What do you mean with the following sentence. It is hard to follow. What has the concentration to do with the composition?          We meant that surface waters along the section were not characterised by high concentration of trace metals. We removed to sentence about composition to make the discussion clearer.

**Conclusion**

This is not a conclusion, it is just a second abstract.          We modified the conclusion.

Line 550: Maybe start the Sentence with: The          Done.

Line 555: The river is not responsible or the PFe at 2500 m depth.          Indeed. We modified the conclusion.

Figure 2 citation: Line 908: Replace "Stations" by "Station"          Done.

Figure 3: Please increase the font size of the ODV graphs, it is almost impossible to read the depth and longitude on a print out.          Done.

Figure 4 caption: Please include that the PFe scale changes within graphs.          Done.

[revised manuscript text omitted]

---

## Author Response (AR4)

Review on "Inputs and processes affecting the distribution of particulate iron in the North Atlantic along the GEOVIDE (GEOTRACES GA01) section" by Gourain et al.

I stopped reviewing the manuscript at line 366. Due to the state of the English language, sentences are difficult, some almost impossible to understand. This needs to be fixed before the next round of review should start. I am suggesting that a native English speaking person has a look and improves the English. Sorry for that, but I need to recommend major revision. However, I am still willing to review the next version, but insist on good English grammar. Sincerely,

Christian Schlosser

Dear Dr Schlosser,

This new version of the manuscript has been proof-read by a native English speaker. Following her advices and suggestions, we modified the structure of many sentences.

You will also find our detailed answers to the specific points your raised below.

We truly hope that you will now find this manuscript suitable for publication.

Kind regards,

Arthur Gourain, on behalf of all coauthors.

Some minor points:

Line 35: Change to "…(0.21 mol mol-1)…"      Done.

Line 38ff: Include the second decimal digit for 0.7 and remove the ratio for the continental crust (you just told it 2 sentences earlier.      Done.

Line 42: Maybe "suspending" would be better than "delivering".      We modified it.

Line 51: Here "deliver" would be better than "bring"      Done.

Line 65ff: What do you mean with dissolution? Dissolution of inorganic PFe? Is not that also part of the regeneration?      Yes, we mean dissolution of inorganic PFe such as iron contained in basaltic grains, and this is now specified in line 67.

From our point of view, the regeneration is the recycling of biogenic PFe.

Line 101: Replace "They" by "Bottles"      Done.

Line 123: Rewrite sentence "before to pass".                Done.

Line 189: Remove "really"                Done.

Line 190ff: What do you mean with "These authigenic particles lead to an enrichment of Mn in particle compositions."

The elemental composition of the particles is driven by the various origins of these particles (lithogenic, biogenic, …). For example, if the particulate bulk is dominated by lithogenic particles, its composition will have a strong imprint of lithogenic elements, e.g. Fe, Al, Ti.

Under certain conditions,  manganese oxides can be generated, with the consequence of depleting dissolved manganese concentrations, and increasing the ambient particulate Mn concentrations. As this statement was indeed confusing, we reorganized the order of sentences in this section (lines 192-200).

Line 195: Replace "of direct and recent" by "for"                Done

Line 226: "modal", do you mean "mode"?        Yes indeed, we changed it.

Line 242: Change "Total particulate concentrations spanned a large range of concentrations from below detection to 304 nmol L-1 for PFe, 1544 nmol L-1 for PAl, 3.5 nmol L-1 for PMn and 402 nmol L-1 for PP."                Done.

Line 275ff: include "depth" after you introduce the depth of a sample , for example "at 25 m depth" in line286. Please apply throughout the paragraph/manuscript.        Done.

Line 299: Remove "candidate"                Done.

Line 300: Replace "ice shelves" by "glaciers"                Done.

Line 350: Remove "candidate"                Done.

Figure 5: Please include in the legend what the different factors are (eg. lithogenic). Makes it easier to follow                Done.

[revised manuscript text omitted]

Figure 3: Left) panel: Distribution of (a) total particulate iron (a, PFe), (b) aluminium (b, PAl), (c) manganese (c, PMn) and (d) phosphorus (d, PP) concentrations (in nmol L$^{-1}$) along the GEOVIDE section. Right panel:) Contribution of the small size fraction (0,45-5 µm) expressed as a percentage (%) of the total concentration of (e) PFe (e), (f) PAl (f), (g) PMn (g) and (h) PP (h). Station IDs and biogeochemical regions are indicated on top of section a. This figure was generated by Ocean Data View (Schlitzer, R., Ocean Data View, odv.awi.de, 2017).

[Figure]

[Figure]

[Figure]

Figure 4: Section of derived contributions of sedimentary inputs (a) manganese bulk sediment proxy (a) and ( b) transmissometry (b) along the GA01 section. Station IDs and biogeochemical region are indicated above the section (a). This figure was generated by Ocean Data View (Schlitzer, R., Ocean Data View, odv.awi.de, 2017).

[Figure]

[Figure]

**Figure 5: Factor fingerprint of the positive matrix factorisation. The four factors are represented in a stacked bar chart**
**of the percentage of variance explained per element.**

Commented [RS1]: It might be a good idea to list what each factor is dominated by, e.g. Factor 1 is dominated by the lithogenic elements, Ti, Fe, Al and Th, etc

[Figure]

**Figure 6: a) Section of the PFe to PAl molar ratio (mol mol⁻¹); (b) contribution of  PFe_litho(%) based on Eq.**
**(1). Station IDs and biogeochemical provinces are indicated above each section. This figure was generated by Ocean**
**Data View (Schlitzer, R., Ocean Data View, odv.awi.de, 2017).**

[Figure]

**Figure 7:** Box and whisker diagram of PFe/PAl molar ratio (mol mol⁻¹) in the different water masses sampled along the GA01 line. Median values for the water masses were as follows: $LSW_{lb}= 0.37$; $LSW_{Ir}=0.44$; $LSW_{WEB}=0.36$; $ISOW_{east}=0.48$; $ISOW_{west}=0.58$; $DSOW_{lab}=0.42$; $DSOW_{Ir}=0.47$; NEADW=0.23; MW=0.22 mol mol⁻¹. Based on their salinity and potential temperature signatures (García-Ibáñez et al., 2015; Figure 2), we applied a Kruskal-Wallis test  molar PFe/PAl ratios of nine water masses (Figure 7) in order to test the presence of significant differences. Water masses for which we had less than 5 data points for PFe/PAl were excluded from this test. As the differences in the median values among the treatment groups were greater than would be expected by chance; the difference in PFe/PAl between water masses is statistically significant (P = <0.001).

[Figure]

[Figure]

**Figure 8: Vertical profiles of (a) PFe (nmol L$^{-1}$, a), (b) lithogenic proportion of particulate iron (PFe$_{litho}$, %, b) and (c) sedimentary proportion of particulate manganese (PMn sediment, %, c) at the Iberian, East-West Greenland and Newfoundland margins.**

[Figure]

**Figure 89: Scatter of the PFe/PAl ratio at the Iberian (red dots), East Greenland (black dots), West Greenland (green**
**dots) and Newfoundland margins (blue dots). Dashed line indicate the UCC ratio (Taylor and McLennan, 1995).**

[Figure]

[Figure]

**Figure 10: Vertical profiles of PFe (nmol L⁻¹, a), lithogenic proportion of particulate iron (%, b) and sedimentary proportion of particulate manganese (%, c) at the Iberian, East-West Greenland and Newfoundland margins.**

[Figure]

Figure 10: PFe total (a); PFe/PAl ratio (b) and beam transmissometry (%) as a function of depth above the seafloor (m) at selected stations where a decrease in transmissometry was recorded.

[Figure]

| | | Fe | Al | P | Mn |
|---|---|---|---|---|---|
| Blank (nmol L$^{-1}$) | 5µm filter | 0.072 | 0.100 | 0.511 | 0.003 |
| | 0.45µm filter | 0.132 | 0.164 | 1.454 | 0.005 |
| Limit of detection (nmol L$^{-1}$) | 5µm filter | 0.011 | 0.030 | 0.365 | 0.001 |
| | 0.45µm filter | 0.026 | 0.046 | 1.190 | 0.001 |
| Recovery CRM (%) | BCR-414 (n=10) | 88 ± 7 | | | 94 ± 7 |
| | MESS-4 (n=5) | 98 ± 14 | 97 ± 14 | 80 ± 30 | 110 ± 18 |
| | PACS-3 (n=8) | 101 ± 9 | 99 ± 14 | 91 ± 34 | 112 ± 11 |

**Table 1: Blank and limit of detection (nmol L$^{-1}$) of the two filters and  certified reference material (CRM) recoveries during GEOVIDE suspended particle digestions.**

| Author | Year | Fraction | Location | Depth range | PFe | PAl | PMn | PP |
|---|---|---|---|---|---|---|---|---|
| This study | | >0.45µm | N. Atlantic (>40°N) | All | bdl-304 | bdl-1544 | bdl-3.5 | bdl-402 |
| Barrett et al. | 2012 | 0.4um | N. Atlantic (25-60°N) | Upper 1000m | 0.29-1.71 | 0.2-19.7 | | |
| Dammshauser et al. | 2013 | >0.2 µm | Eastern tropical N.A. | 0-200 | | | 0.59-17.7 | |
| Dammshauser et al. | 2013 | >0.2 µm | Meridional Atlantic | 0-200 | | | 0.35-16.1 | |
| Lam et al. | 2012 | 1–51 um | Eastern tropical N.A. | 0-600 | ND-12 | | | |
| Lannuzel et al. | 2011 | >0.2 µm | East Antarctic | Surface | | 0.02-10.67 | 0.01-0.14 | |
| Lannuzel et al. | 2014 | >0.2 µm | East Antarctic | Fast ice | 43-10385 | 121-31372 | 1-307 | |
| Lee et al. | 2017 | >0.8 µm | Eastern tropical S.Pacific | All | bdl-159 | bdl-162 | bdl-8.7 | bdl-983 |
| Marsay et al. | 2017 | >0.4 µm | Ross Sea | All | 0.68-57.3 | ND-185 | ND-1.4 | 5.4-404 |
| Milne et al. | 2017 | >0.45µm | Sub-tropical N.A. | All | ND-140 | ND-800 | | |
| Ohnemus et al. | 2015 | 0.8–51 µm | N. Atlantic | All | 0-938 | 0-3600 | | |
| Planquette et al. | 2009 | >53 µm | Southern Ocean | 30-340 | 0.15–13.2 | 0.11–25.5 | | |
| Schlosser et al. | 2017 | >1 µm | South Georgia Shelf | All | 0.87-267 | 0.6-195 | 0.01-3.85 | |
| Sherrell et al. | 1998 | 1-53um | Northeast Pacific | 0-3557 | | 0.0-54.2 | | |
| Weinstein et al. | 2004 | >53 µm | Labrador Sea | 0-250 | 0.1-1.2 | 0.1-1.5 | | |
| Weinstein et al. | 2004 | 0.4– 10um | Labrador Sea | 0-250 | 2.5 | 3.6 | 0.05 | |
| Weinstein et al. | 2004 | >0.4 µm | Gulf of Maine | 0-300 | 34.8 | 109 | | |

**Table 2: Concentration (in nmol L$^{-1}$) of trace elements (PFe, Pal, PMn and PP) in suspended particles collected in diverse regions of the world's ocean. Bdl: below detection limit, ND: non-determined.**

---

## Author Response (AR5)

Review on "Inputs and processes affecting the distribution of particulate iron in the North
Atlantic along the GEOVIDE (GEOTRACES GA01) section" by Gourain et al.
The manuscript has improved much! I agree with the scientific findings and suggestions
made, but still the language is sometimes hard to follow and paragraphs are not always, but
sometimes very long. All this makes it to a real challenge to read the manuscript.
Dear Dr Schlosser,
We shortened some of the paragraphs to make it more readable as suggested. You will also
find our detailed answers to the specific points your raised below. We also joined the
manuscript with track change.
We truly hope that you will now find this manuscript suitable for publication.
Kind regards,
Arthur Gourain, on behalf of all coauthors.
From the scientific part, I have just some minor points, listed below
Line 32: Insert "thermohaline overturning circulation."          Done.
Line 131: I do not know what the Berger citation has to do with your results. I would remove
the Berger et al. citation!          We rephrase this sentence: "…while the other half was archived at -
20 °C for SEM analyses or acid leaching of "labile" metals following Berger et al. (2008) method (to be
published separately).". Line 132.
Line 210: Include "Close to the sea floor…."Done.
Line 318: Replace (PMF, factor 3 = 4.1%)          Done.
Line 319ff: Replace ", but its contribution is most likely obscured…"          Done.
Line 321ff: The reasoning is plausible, but please shorten this paragraph, it is too long. You
can distill the message down to a couple sentences.  We shortened the paragraph. From line
322 to line 335.
Line 388: Replace "in" by "to the Atlantic Ocean."  Done.
Line 401-409: I do not follow the reasoning! PP is high because of high DFe and elevated
primary productivity, which subsequently reduced DFe by biological utilization. First you
need to show that DFe concentrations are elevated. Then use Chl a to show that the
production is elevated, and then you may argue that the high Pfe with a high PFe/Pal ratio
comes from elevated productivity.  Values of DFe and Chl-a have been added to the
sentence. Line 399ff.
Line 410ff: Shorten the paragraph, you say a lot without saying something useful. We kept
this paragraph as it was. In this paragraph, we discuss the dispersion of the shelf signal for the
first and only time. We think it is important to keep it as it was.

Line 446ff: This paragraph is out of context. Also the next one, which I suppose was written to sum up the results. But somehow it doesn't, maybe better to create another section.
We deleted this paragraph.

Line 466ff: What is new, that has not been said earlier. Nepheloid layers have been already introduced!    The introductory paragraph is saying for the first time, the importance of BNLs along GEOVIDE and their differences between each other. We think keeping this paragraph here is important.

Line 490ff: You could also use DFe and DMn as a tracer. What are these elements showing! DFe and DMn were not showing any elevated concentration above the ridge.

Line 527ff: This is speculative!    From your previous comments, we modify the paragraph Line 399ff in order to demonstrate this statement.

[revised manuscript text omitted]